# Probabilistic inverse optimal control for non-linear partially observable systems disentangles perceptual uncertainty and behavioral costs

**Dominik Straub** *
Centre for Cognitive Science
Technische Universität Darmstadt
dominik.straub@tu-darmstadt.de

**Matthias Schultheis** *
Centre for Cognitive Science
Technische Universität Darmstadt
matthias.schultheis@tu-darmstadt.de

**Heinz Koeppl**
Centre for Cognitive Science
Technische Universität Darmstadt
heinz.koeppl@tu-darmstadt.de

**Constantin A. Rothkopf**
Centre for Cognitive Science
Technische Universität Darmstadt
constantin.rothkopf@tu-darmstadt.de

## Abstract

Inverse optimal control can be used to characterize behavior in sequential decision-making tasks. Most existing work, however, is limited to fully observable or linear systems, or requires the action signals to be known. Here, we introduce a probabilistic approach to inverse optimal control for partially observable stochastic non-linear systems with unobserved action signals, which unifies previous approaches to inverse optimal control with maximum causal entropy formulations. Using an explicit model of the noise characteristics of the sensory and motor systems of the agent in conjunction with local linearization techniques, we derive an approximate likelihood function for the model parameters, which can be computed within a single forward pass. We present quantitative evaluations on stochastic and partially observable versions of two classic control tasks and two human behavioral tasks. Importantly, we show that our method can disentangle perceptual factors and behavioral costs despite the fact that epistemic and pragmatic actions are intertwined in sequential decision-making under uncertainty, such as in active sensing and active learning. The proposed method has broad applicability, ranging from imitation learning to sensorimotor neuroscience.

## 1 Introduction

Inverse optimal control (IOC) is the problem of inferring an agent's cost function and other properties of their internal model from behavior. While IOC has been a fundamental task in artificial intelligence and machine learning, particularly reinforcement learning (RL) and robotics, it has widespread applicability in several scientific fields including behavioral economics, psychology, and neuroscience. For example, in cognitive science and sensorimotor neuroscience, optimal control models have explained key properties of behavior, such as speed-accuracy trade-offs [1] or the minimum intervention principle [2]. But, while researchers usually build an optimal control model and compare its predictions to behavior, certain parameters of the agent's internal processes are typically unknown. For example, an agent might have uncertainty about their perception or experience intrinsic costs of behavior. These parameters are different between individuals and inferring them from observed behavior can help to understand internal tradeoffs between behavioral goals, perceptual and cognitive

---

*equal contribution

37th Conference on Neural Information Processing Systems (NeurIPS 2023).

processes, and predict behavior under novel conditions. Applying IOC in these domains poses several challenges that make most previous methods not viable.

First, most IOC methods assume the agent's control signals to be known. This assumption, while convenient in simulations or robotics, where the control signals may be easily quantified, does not hold in many other real-world applications. In transfer learning or behavioral experiments, the control signals are internal quantities of an animal or human, e.g., neural activity or muscle activations, and are therefore not straightforwardly observable. Thus, we consider the scenario where a researcher has observations of the system's state only, i.e., measurements of behavior.

Second, most IOC methods model the variability of the agent using a stochastic policy in a maximum causal entropy formulation [MCE; 3]. Behavioral variability in biological systems, however, is known to arise from multiple distinct sources [4]. There is noise in the sensory system, which makes the state of the world partially observable, and in the motor system. In sensorimotor neuroscience, the uncertainty in the sensory and motor systems can be characterized quantitatively by formulating accurate models, which are helpful to understand behavioral variability [5].

Third, many IOC methods are based on matching feature expectations of the cost function between the model and data [3], and are thus not easily adapted to infer other model parameters. In a behavioral experiment, however, researchers are often interested in inferring the noise characteristics of the sensorimotor system or other properties of the agent's internal model besides the cost function [6].

Fourth, while the theory of linear-quadratic-Gaussian (LQG) control [7] is suited to deal with the issues above, many real-world systems are not captured by the LQG assumptions. First, the dynamics may be non-linear, e.g., in robotics and motor control when controlling joint angles in a kinematic chain. Second, the variability of the system may not be captured by normal distributions, e.g., in sensorimotor control, where the standard deviation of sensory and control signals scales with their means. While iterative methods for solving the optimal control problem such as iterative variants of LQG [8] exist, here we consider the corresponding inverse problem.

To address these issues, we adopt a probabilistic perspective. We distinguish between the control problem faced by the agent and the IOC problem faced by the researcher. From the agent's perspective, the problem consists of acting in a partially observable Markov decision process (POMDP; Fig. 1 A). We consider the setting of continuous states and controls, stochastic non-linear dynamics, partial observations, and finite horizon. For this setting, there are efficient approximately optimal solutions to the estimation and control problem (see Section 2). The researcher, on the other hand, is interested in inferring properties of the agent's model and cost function. The IOC problem from their perspective can be formulated using a probabilistic graphical model (Fig. 1 B), in which the state of the system is observed, while variables internal to the agent are latent.

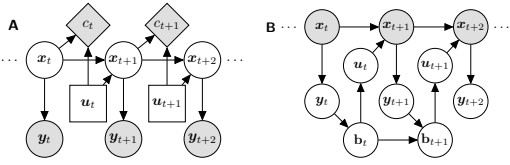

Figure 1: **A** Decision network from the agent's perspective [notation from 9]. At each time step, the agent receives a noisy observation $\boldsymbol{y}_t$ of the state $\boldsymbol{x}_t$, performs a control $\boldsymbol{u}_t$, and incurs a cost $c_t$. **B** Probabilistic graphical model from the researcher's perspective, who observes a trajectory $\boldsymbol{x}_{1:T}$ of an agent. Quantities internal to the agent, i.e. observations $\boldsymbol{y}_t$, beliefs $\mathbf{b}_t$ and control signals $\boldsymbol{u}_t$, are not directly observed.

Here, we unify MCE models, which are agnostic regarding the probabilistic structure causing the observed stochasticity of the agent's policy, with IOC methods that involve an explicit observation model. First, we define the IOC problem where we allow for both: We employ an explicit observation model, but also allow the agent to have additional stochasticity through an MCE policy. Second, we provide a solution to the IOC problem in this setting by approximate filtering of the agent's state estimate via local linearization, which allows marginalizing over these latent variables and deriving an approximate likelihood function for observed trajectories given parameters (Section 3). By maximizing the approximate likelihood function, an estimate of the optimal parameters can be determined. Third, we evaluate our proposed method on two classic control tasks, pendulum and cart pole, and on two human behavioral tasks, navigation and a manual reaching (Section 4.2 - Section 4.3). Fourth, we show that our approach allows disentangling the influences of perceptual uncertainty and behavioral costs on information-seeking behavior in the light-dark domain (Section 4.4).

**Related work**

Inferring costs or utilities from behavior has been of interest for a long time in several scientific fields, such as behavioral economics, psychology, and neuroscience [10–12]. More specific to the problem formulation adopted here, estimating objective functions in the field of control was first investigated by Kalman [13] in the context of deterministic linear systems with quadratic costs. More recent formulations were developed first for discrete state and control spaces under the term inverse reinforcement learning [IRL; 14, 15], including formulations allowing for stochasticty in action selection [16]. In this line, the maximum entropy [ME; 17] and MCE formulation [3] gave rise to many new methods, e.g., for non-linear continuous systems via linearization [18] or importance sampling [19] for fully observable deterministic systems.

IOC methods for stochastic systems have been developed in the setting of affine control dynamics [20, 21]. Arbitrary non-linear stochastic dynamics in the infinite horizon setting have been approached using model-free deep MCE IRL [22, 23]. The latter approaches, however, do not yield interpretable representations, as the cost function is represented by a neural network. Further, past methods based on MCE are limited to estimating cost functions and cannot be used to infer other latent quantities, such as noises or subjective beliefs. The partially observable setting for IOC has previously been addressed for discrete state-action spaces [24] and continuous states with discrete actions [25]. Schmitt et al. [26] addressed systems with linear dynamics and continuous controls for a linear switching observation model. Other work has considered partial observability from the researcher's perspective, e.g., through occlusions [27, 28]. There are some IOC methods which are applicable to partially observable stochastic systems: In our previous work [29] we regarded LQG systems, while the work of Chen and Ziebart [30] can be used to estimate cost functions that depend on the state only. Non-linear dynamics in the infinite-horizon setting and the joint estimation of model parameters have been approached by Kwon et al. [31] by training a policy network as a function of the whole parameter space. This work, however, also assumes the control signals to be given and a stationary, deterministic policy.

Applications of IOC methods range from human locomotion [32] over spatial navigation [33], table tennis [34], to attention switching [35], and target tracking [36]. Other work has been aimed at inferring other properties of control tasks, e.g., the dynamics model [6], learning rules [37], or discount functions [38]. Several subfields of robotics including imitation and apprenticeship learning [39] as well as transfer learning [40] have also employed IOC.

## 2 Background

Before we introduce our probabilistic approach to inverse optimal control, we give an overview of the control and filtering problems faced by the agent and algorithms that can be used to solve it. For a summary of our notation in this paper, see Appendix A.

### 2.1 Partially observable Markov decision processes (POMDPs)

We consider a special case of POMDPs [41, 42], a discrete-time stochastic non-linear dynamical system (Fig. 1 A) with states $\boldsymbol{x}_t \in \mathbb{R}^n$ following the dynamics equation $\boldsymbol{x}_{t+1} = f(\boldsymbol{x}_t, \boldsymbol{u}_t, \boldsymbol{v}_t)$, where $f$ is the dynamics function, $\boldsymbol{u}_t \in \mathbb{R}^u$ are the controls and $\boldsymbol{v}_t \sim \mathcal{N}(0, I)$ is $v$-dimensional Gaussian noise. We assume that the agent has only partial observations $\boldsymbol{y}_t \in \mathbb{R}^m$, following $\boldsymbol{y}_t = h(\boldsymbol{x}_t, \boldsymbol{w}_t)$, with $h$ the stochastic observation function and $\boldsymbol{w}_t \sim \mathcal{N}(0, I)$ $w$-dimensional Gaussian noise. While $\boldsymbol{v}_t$ and $\boldsymbol{w}_t$ are standard normal random variables, the system can incorporate general control- and state-dependent noises through non-linear transformations within the dynamics function $f$ and observation function $h$. The agent's goal is to minimize the expected cost over a time horizon $T \in \mathbb{N}$, defined by

$$J = \mathbb{E}\left[c_T(\boldsymbol{x}_T) + \sum_{t=1}^{T-1} c_t(\boldsymbol{x}_t, \boldsymbol{u}_t)\right],$$

consisting of a final state cost $c_T(\boldsymbol{x}_T)$ and a cost at each time step $c_t(\boldsymbol{x}_t, \boldsymbol{u}_t)$.

## 2.2 Iterative linear-quadratic Gaussian (iLQG)

The control problem from Section 2.1 can be solved approximately using iLQG [8, 43]. This method iteratively linearizes the dynamics and quadratizes the costs around a nominal trajectory, $\{\bar{\boldsymbol{x}}_i, \bar{\boldsymbol{u}}_i\}_{i=1,...,T}$, with $\bar{\boldsymbol{x}}_i \in \mathbb{R}^n, \bar{\boldsymbol{u}}_i \in \mathbb{R}^u$, and computes the optimal linear control law, $\boldsymbol{u}_t = \pi_t(\boldsymbol{x}_t) = L_t(\boldsymbol{x}_t - \bar{\boldsymbol{x}}_t) + \mathbf{m}_t + \bar{\boldsymbol{u}}_{1:T}$ for the approximated system. The quantities $L_t$ and $\mathbf{m}_t$ are the control gain and offset, respectively, and determined through a backward pass for the current reference trajectory. In the following iteration, the determined optimal control law is used to generate a new reference trajectory and the process is repeated until the controller converges.

## 2.3 Maximum causal entropy (MCE) reinforcement learning

The goal of MCE RL is to minimize the expected cost as in Section 2.2, while maximizing the conditional entropy of the stochastic policy $\Pi_t(\boldsymbol{u}_t \mid \boldsymbol{x}_t)$, i.e., to minimize $\mathbb{E}[J(\boldsymbol{x}_{1:T}, \boldsymbol{u}_{1:T}) - \sum_{t=1}^{T-1} \mathcal{H}(\Pi_t(\boldsymbol{u}_t \mid \boldsymbol{x}_t))]$. This formulation has been used to treat RL as probabilistic inference [44–46] and model the stochasticity of the agent in IRL [17, 3]. The objective of IRL is to maximize the likelihood of states and controls $\{\boldsymbol{x}_t, \boldsymbol{u}_t\}_{t=1,...,N}$, induced by the maximum entropy policy. The resulting optimal policy is given by the distribution $\Pi_t(\boldsymbol{u}_t \mid \boldsymbol{x}_t) = \exp(Q_t(\boldsymbol{x}_t, \boldsymbol{u}_t) - V_t(\boldsymbol{x}_t))$, where $Q_t$ is the soft Q-function and $V_t$ the normalization [47]. For general dynamics and reward functions, it is not feasible to compute the soft Q-function exactly. Approximate solutions have been derived using linearization [18], importance sampling [19], or deep function approximation [23]. For linear dynamics and quadratic costs, the optimal policy is a Gaussian distribution $\Pi_t(\boldsymbol{u}_t \mid \boldsymbol{x}_t) = \mathcal{N}(\boldsymbol{u}_t; L_t\boldsymbol{x}_t, -H_t^{-1})$, where $L_t$ and $H_t$ result from the optimal LQG controller [48]. More detailed formulas are provided in Appendix B.

## 2.4 Extended Kalman filter (EKF)

Given the system defined in Section 2.1, the optimal filtering problem is to compute a belief distribution of the current state given past observations, i.e., $p(\mathbf{x}_t \mid \mathbf{y}_{1:t-1})$. For linear-Gaussian systems, the solution is given in closed form and known as the Kalman filter [49]. In case of non-linear systems as in Section 2.1, a Gaussian approximation to the optimal belief can be computed using the extended Kalman filter (EKF) via $\mathbf{b}_{t+1} = f(\mathbf{b}_t, \boldsymbol{u}_t, 0) + K_t(\boldsymbol{y}_t - h(\mathbf{b}_t, 0))$, where $\mathbf{b}_t \in \mathbb{R}^n$ denotes the mean of the Gaussian belief $p(\boldsymbol{x}_t \mid \boldsymbol{y}_1, \ldots, \boldsymbol{y}_{t-1})$. The matrix $K_t$ denotes the Kalman gain for time $t$ and is computed by applying the Kalman filter to the system locally-linearized around the nominal trajectory obtained by the approximate optimal control law of iLQG (Section 2.2).

# 3 Probabilistic inverse optimal control

We consider an agent acting in a partially observable Markov decision process (POMDP) as introduced in Section 2.1. We assume that the agent acts at time $t$ based on their belief $\mathbf{b}_t$ about the state of the system $\boldsymbol{x}_t$, which evolves according to $\mathbf{b}_{t+1} = \beta_t(\mathbf{b}_t, \boldsymbol{u}_t, \boldsymbol{y}_t)$. While the belief of the agent is defined commonly as a distribution over the true state, here we model $\mathbf{b}_t$ as a finite-dimensional summary statistics of the distribution, i.e., $\mathbf{b}_t \in \mathbb{R}^b$. The function $\beta_t : \mathbb{R}^b \times \mathbb{R}^u \times \mathbb{R}^m \to \mathbb{R}^b$ is called belief dynamics. We further assume that the agent follows a time-dependent policy $\pi_t : \mathbb{R}^b \times \mathbb{R}^j \to \mathbb{R}^u$, i.e., $\boldsymbol{u}_t = \pi_t(\mathbf{b}_t, \boldsymbol{\xi}_t)$, which can be stochastic with $\boldsymbol{\xi}_t \sim \mathcal{N}(0, I)$.

In the inverse optimal control problem, the goal is to estimate parameters $\boldsymbol{\theta} \in \mathbb{R}^p$ of the agent's optimal control problem given the model and trajectory data. These parameters can include properties of the agent's cost function, the sensory and control systems of the agent, or the system's dynamics. More precisely, we assume that we are given a parametric form of the system dynamics, belief dynamics, cost function, and initial belief of the agent, which all might depend on the unknown parameters that we aim to infer. Further we assume a set of state trajectories. Importantly, we do not assume knowledge of the agent's belief. We follow a probabilistic approach to inverse optimal control, i.e., we consider the likelihood function

$$p(\boldsymbol{x}_{1:T} \mid \boldsymbol{\theta}) = p(\boldsymbol{x}_1 \mid \boldsymbol{\theta}) \prod_{t=1}^{T-1} p(\boldsymbol{x}_{t+1} \mid \boldsymbol{x}_{1:t}, \boldsymbol{\theta}), \tag{1}$$

describing the probability of the observed trajectory data $\boldsymbol{x}_{1:T} := \{\boldsymbol{x}_1, \ldots, \boldsymbol{x}_T\}$ given the parameters. For a set of trajectories, we assume them to be independent given the parameters, so that the likelihood factorizes into single trajectory likelihoods of the form in Eq. (1). In this equation, generally, each state $\boldsymbol{x}_{t+1}$ depends on all previous states $\boldsymbol{x}_1, \ldots, \boldsymbol{x}_t$, because the agent's internal noisy observations and control signals are not accessible to the researcher (Fig. 1 B). Therefore, the Markov property does not hold from the researcher's perspective, rendering computation of the likelihood function intractable. To deal with this problem, we employ two key insights: First, the joint dynamical system of the states and the agent's belief is Markovian [50]. Second, by keeping track of the distribution over the agent's belief, i.e., by performing belief tracking [29], we can iteratively compute the individual factors of the likelihood function in Eq. (1). In our IOC method, the goal is to maximize the likelihood w.r.t. the parameters $\boldsymbol{\theta}$. To do so, we use gradient-based optimization with automatic differentiation to differentiate through the likelihood for computing the optimal parameters (Algorithm 1). An implementation of our algorithm is publicly available[2].

We first introduce a general formulation of the IOC likelihood involving marginalization over the agent's internal beliefs in Section 3.1. Then, we show how to make the computations tractable by local linearization in Section 3.2. In Section 3.3, we provide details for suitable linearization points, which enables us to evaluate the approximate likelihood within a single forward pass.

## 3.1 Likelihood formulation

We start by defining a joint dynamical system of states and beliefs [50], in which each depends only on the state and belief at the previous time step and the noises. For that, we insert the policy into the dynamics and the policy and observation function into the belief dynamics, yielding the equation

$$\begin{bmatrix} \boldsymbol{x}_{t+1} \\ \mathbf{b}_{t+1} \end{bmatrix} = \begin{bmatrix} f(\boldsymbol{x}_t, \pi_t(\mathbf{b}_t, \boldsymbol{\xi}_t), \boldsymbol{v}_t) \\ \beta_t(\mathbf{b}_t, \pi_t(\mathbf{b}_t, \boldsymbol{\xi}_t), h(\boldsymbol{x}_t, \boldsymbol{w}_t)) \end{bmatrix} =: g(\boldsymbol{x}_t, \mathbf{b}_t, \boldsymbol{v}_t, \boldsymbol{w}_t, \boldsymbol{\xi}_t). \tag{2}$$

For given values of $\boldsymbol{x}_t$ and $\mathbf{b}_t$, this equation defines the distribution $p(\boldsymbol{x}_{t+1}, \mathbf{b}_{t+1} \mid \boldsymbol{x}_t, \mathbf{b}_t)$, as $\boldsymbol{v}_t, \boldsymbol{w}_t, \boldsymbol{\xi}_t$ are independent of $\boldsymbol{x}_{t+1}$ and $\mathbf{b}_{t+1}$. Importantly, with this formulation, the control signals are only implicitly regarded through the policy, as we assume them to be latent for the researcher. In Section 3.2 we will introduce an approximation via linearization, which leads to a closed-form expression for $p(\boldsymbol{x}_{t+1}, \mathbf{b}_{t+1} \mid \boldsymbol{x}_t, \mathbf{b}_t)$.

One can use this Markovian joint dynamical system to compute the likelihood factors for each time step [29]. To this end, we first rewrite the individual likelihood terms $p(\boldsymbol{x}_{t+1} \mid \boldsymbol{x}_{1:t})$ of Eq. (1) by marginalizing over the agent's belief at each time step, i.e.,

$$p(\boldsymbol{x}_{t+1} \mid \boldsymbol{x}_{1:t}) = \int p(\boldsymbol{x}_{t+1}, \mathbf{b}_{t+1} \mid \boldsymbol{x}_{1:t}) \, \mathrm{d}\mathbf{b}_{t+1}. \tag{3}$$

As the belief is an internal quantity of the agent and thus not observable to the researcher, we keep track of its distribution, $p(\mathbf{b}_t \mid \boldsymbol{x}_{1:t})$. For this, we rewrite

$$p(\boldsymbol{x}_{t+1}, \mathbf{b}_{t+1} \mid \boldsymbol{x}_{1:t}) = \int p(\boldsymbol{x}_{t+1}, \mathbf{b}_{t+1} \mid \boldsymbol{x}_t, \mathbf{b}_t) \, p(\mathbf{b}_t \mid \boldsymbol{x}_{1:t}) \, \mathrm{d}\mathbf{b}_t, \tag{4}$$

where we have exploited the fact that the joint dynamical system of states and beliefs is Markovian. The distribution $p(\mathbf{b}_t \mid \boldsymbol{x}_{1:t})$ acts as a summary of the past states and can be computed by conditioning on the current state, i.e.,

$$p(\mathbf{b}_t \mid \boldsymbol{x}_{1:t}) = \frac{p(\boldsymbol{x}_t, \mathbf{b}_t \mid \boldsymbol{x}_{1:t-1})}{p(\boldsymbol{x}_t \mid \boldsymbol{x}_{1:t-1})}. \tag{5}$$

After determining $p(\mathbf{b}_t \mid \boldsymbol{x}_{1:t})$, we can propagate it through the joint dynamical system to arrive at the distribution $p(\boldsymbol{x}_{t+1}, \mathbf{b}_{t+1} \mid \boldsymbol{x}_{1:t})$. To obtain the belief distribution of the following time step, $p(\mathbf{b}_{t+1} \mid \boldsymbol{x}_{1:t+1})$, we condition on the observed state $\boldsymbol{x}_{t+1}$. To obtain the likelihood contribution, on the other hand, we marginalize out $\mathbf{b}_{t+1}$. To summarize, starting with an initialization $p(\mathbf{b}_0)$, we can compute the individual terms $p(\boldsymbol{x}_{t+1}|\boldsymbol{x}_{1:t})$ of the likelihood by executing Algorithm 1 (Appendix C).

---

[2]`https://github.com/RothkopfLab/nioc-neurips`

## 3.2 Tractable likelihood via local linearization

While the marginalization and propagating operations in the previous section can be done in closed form for linear-Gaussian systems, this is no longer feasible for non-linear systems. Therefore, we follow the approach of local linearization used in iLQG (Section 2.2) and the EKF (Section 2.4). For the belief statistics, we consider the mean of the agent's belief, i.e., $\mathbf{b}_t = \mathbb{E}[\boldsymbol{x}_t \mid \boldsymbol{y}_1, \ldots, \boldsymbol{y}_{t-1}]$ and initialize the distribution for the first time step as a Gaussian, $p(\mathbf{b}_1) = \mathcal{N}(\mu_1^{(b)}, \Sigma_1^{(b)})$. We then approximate the distribution $p(\boldsymbol{x}_{t+1}, \mathbf{b}_{t+1} \mid \boldsymbol{x}_t, \mathbf{b}_t)$ as a Gaussian by applying a first-order Taylor expansion of $g$.

To obtain a closed-form expression for $g$, which we can linearize, we model the agent's policy using iLQG (Section 2.2) and the belief dynamics using the EKF (Section 2.4). This choice leads to an affine control and belief given $\mathbf{b}_t$, making linearization of $p(\boldsymbol{x}_{t+1}, \mathbf{b}_{t+1} \mid \boldsymbol{x}_t, \mathbf{b}_t)$ straightforward. To allow for additional stochasticity in the agent's policy, we use the MCE formulation (Section 2.3). For linearized dynamics, the MCE policy is given by a Gaussian distribution, so that $\pi_t(\mathbf{b}_t, \boldsymbol{\xi}_t) = L_t(\mathbf{b}_t - \bar{\boldsymbol{x}}_{1:T}) + \mathbf{m}_t + \bar{\boldsymbol{u}}_{1:T} - \tilde{H}_t \boldsymbol{\xi}_t$, with $\tilde{H}_t$ the Cholesky decomposition of $H_t$, and can be marginalized out in closed form.

The approximations we have introduced allow us to solve the integral in Eq. (4) in closed form by applying standard equations for linear transformations of Gaussians, resulting in

$$p(\boldsymbol{x}_{t+1}, \mathbf{b}_{t+1} \mid \boldsymbol{x}_{1:t}) \approx \mathcal{N}(\mu_t, \Sigma_t),\tag{6}$$

with $\mu_t = g(\boldsymbol{x}_t, \mu_t^{(b)}, 0, 0, 0)$ and $\Sigma_t = \mathbb{J}_\mathbf{b} \Sigma_t^{(b)} \mathbb{J}_\mathbf{b}^T + J_{\boldsymbol{v}} \mathbb{J}_{\boldsymbol{v}}^T + \mathbb{J}_{\boldsymbol{w}} \mathbb{J}_{\boldsymbol{w}}^T + \mathbb{J}_{\boldsymbol{\xi}} \mathbb{J}_{\boldsymbol{\xi}}^T$, where $\mathbb{J}_\bullet$ denotes the Jacobian of $g$ w.r.t. $\bullet$, evaluated at $(\boldsymbol{x}_t, \mu_t^{(b)}, 0, 0, 0)$. Under this Gaussian approximation, both remaining operations of Algorithm 1 (Appendix C) can also be performed in closed form. A more detailed derivation and representation of these formulas can be found in Appendix D. If the agent has full observations of the system's state, the inverse optimal control problem is simplified significantly (see Appendix E). Details about the implementation are provided in Appendix F.

## 3.3 Data-based linearization

The forward optimal control problem is commonly solved by starting with a randomly initialized nominal trajectory and iterating between computing the locally optimal control law and linearization until convergence. To compute the likelihood in the inverse problem, we can take a more efficient approach by linearizing directly around the given trajectory $\boldsymbol{x}_{1:T}$. We then need to perform only one backward pass to compute an approximately optimal control law given the current parameters, and a forward pass to compute an approximately optimal filter. This, in particular, allows efficient computation of the gradient of the likelihood function for the optimization procedure. As we assume the controls to be unobservable, but they are needed for the linearization, we compute estimates of the controls by minimizing the squared difference of the noiseless predicted states and the actual states (see Appendix G). Note that these estimated controls are only used for the linearization, but are not used as observed controls in the IOC likelihood itself. In the case where the full state is not observable, we cannot linearize around the trajectory. For these cases, we propose two approaches to compute gradients based on implicit differentiation and differentiating only through the last iteration. As this setting is not the main focus of this paper, details of these approaches are provided in Appendix H.

## 4 Experiments

We evaluated our method on two classic control tasks, i.e., Pendulum and Cart Pole, and two human behavioral tasks, manual reaching and navigation. To evaluate the accuracy of the parameter estimates obtained by our method and to compare it against a baseline, we computed absolute relative errors per parameter, i.e., $|(\theta - \hat{\theta})/\theta|$. This metric makes averages across parameters on different scales more interpretable compared to other metrics such as root mean squared errors. For each task, we simulated 100 sets of parameters from a uniform distribution in logarithmic space. For each set of parameters, we simulated 50 trajectories. We then maximized the log likelihood using gradient-based optimization with automatic differentiation [L-BFGS algorithm; 51]. See Appendix I for a summary of the hyperparameters of our experiments.

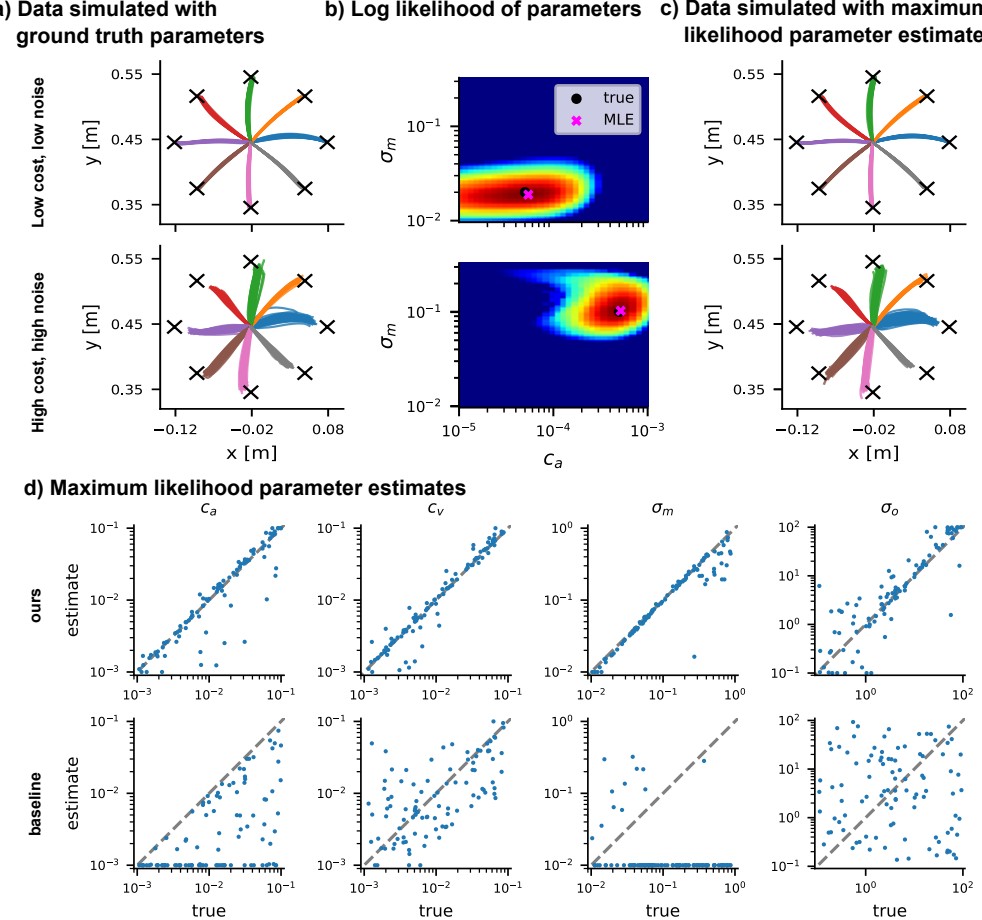

**a) Data simulated with ground truth parameters**

**b) Log likelihood of parameters**

**c) Data simulated with maximum likelihood parameter estimates**

**d) Maximum likelihood parameter estimates**

Figure 2: **IOC for non-linear reaching. A** Simulated trajectories for eight targets. Increasing the control cost and the motor noise affects the trajectories, since reaching the target becomes less important and variability increases. **B** IOC log likelihood for control cost $c_a$ and motor noise $\sigma_m$. The maximum likelihood estimate (pink cross) is close to the ground truth parameters (black dot). **C** Simulated trajectories using the MLEs from B. The simulations are visually indistinguishable from the ground truth data. **D** True parameters plotted against maximum likelihood estimates. Top row: our method, bottom row: MCE baseline. The columns contain the four different model parameters (control cost $c_a$, velocity cost $c_v$, motor noise $\sigma_m$, observation noise $\sigma_o$).

All tasks we consider have four free parameters: control cost $c_a$, cost of final velocity $c_v$, motor noise $\sigma_m$, and observation noise $\sigma_o$ of the agent. In the fully observable case, we leave out the observation noise parameter and only infer the three remaining parameters. For concrete definitions of the parameters in each specific task, see Appendix J.

## 4.1 Baseline method

For a comparison to previously proposed methods, we applied a baseline method based on the maximum causal entropy (MCE) approach [3]. As for this approach, control signals of the observed trajectories are required, we use the estimates of the controls that we determine in our proposed method for the data-based linearization (Section 3.3). Note that the baseline, representative for applicable past IOC methods, does not have an explicit model of partial observability. Further note that past methods based on MCE are limited to estimating cost functions, so that parameters such as the agent's noise cannot be inferred. For the specific MCE linearization-based baseline we consider, it is actually straightforward to maximize the likelihood with respect to the noise parameters, which enables us to evaluate noise estimates. To show that this approach constitutes a suitable baseline, in

Appendix K.3, we provide results for the case where the true control signals are known and there is no partial observability. More details of the baseline are provided in Appendix B.

## 4.2 Evaluation on manual reaching task

We evaluate the method on a reaching task with a non-linear two-joint biomechanical arm model, which has been applied to reaching movements in the sensorimotor neuroscience literature [e.g., 52, 53]. The agent's goal is to move its arm towards a target at position $\mathbf{e}^\star$ by controlling the torque to the two joints. This objective is expressed as a non-quadratic cost function of the joint angles,

$$
J = \|\mathbf{e}_T - \mathbf{e}^*\|^2 + c_v \|\dot{\mathbf{e}}_T\|^2 + c_a \sum_{t=1}^{T-1} \|\mathbf{u}_t\|^2 , \tag{7}
$$

since the final position and velocity of the hand $\mathbf{e}_T, \dot{\mathbf{e}}_T$ are non-linear functions of the joint angles. See Appendix J.1 for details.

We use a fully observable [8] and a partially observable version of the task [43]. Fig. 2 A shows simulations from the model with two different parameter settings. Evaluating the likelihood function for a grid of two of the parameters (Fig. 2 B) confirms that it has its maxima close to the true parameter values. Simulated data using the maximum likelihood estimates look indistinguishable from the ground truth data (Fig. 2 C).

In Fig. 2 D, we show maximum likelihood estimates and true values for repeated runs with different random parameter settings. The parameter estimates of our method closely align with the true parameter values, showing that we can successfully recover the parameters from data. The baseline method, in contrast, shows considerably worse performance, in particular for estimating noises due to the lacking explicit representation of partial observability. Importantly, even when the true control signals are provided, the noise parameter estimates of the baseline are not well estimated (Appendix K.3). Estimates for the fully observable case are provided in Appendix K.2. The median absolute relative errors of our method were 0.11, while they were 0.93 for the baseline. The influence of missing control signals and of the lack of an explicit observation model in the baseline can be observed by comparing the results to the fully observable case and the case of given control signals in Appendix K.2 and Appendix K.3.

## 4.3 Quantitative evaluation on other tasks

To show that our method works for a range of different tasks, we evaluated it on the three other tasks (navigation, pendulum and cart pole). In the navigation task, we consider an agent navigating to a target under non-linear dynamics while receiving noisy observations from a non-linear observation model. To reach the target, the agent can control the angular velocity of their heading direction and the acceleration with which they move forward. The agent observes noisy versions of the distance to the target and the target's bearing angle. We provide more details about the experiment in Appendix J.2. Maximum likelihood parameter estimates for the navigation task are shown for the partially observable case in Fig. S6 and for the fully observable case in Fig. S10. As for the reaching task, our method provides parameter estimates close to the true ones, while the estimates of the baseline deviate for many trials. Median absolute relative errors of our method were 0.31, while they were 1.99 for the baseline (Fig. 3).

The two classic control tasks (Pendulum and Cart Pole) are based on the implementations in the gym library [54]. Because these tasks are neither stochastic nor partially observable in their standard formulations, we introduce noise on the dynamics and turn them into partially observable problems by defining a stochastic observation function (see Appendix J.3). In Appendix K, we show the parameter estimates for the Pendulum (Fig. S4) and for the Cart Pole (Fig. S5) for the partially observable case, while Fig. S8 and Fig. S9 show the fully observable case, respectively. One can observe that the results are qualitatively similar to the ones in the reaching and navigation tasks, showing that our method provides accurate estimates of the parameters. Median absolute relative errors of our method were 0.12 and 0.41, while for the baseline they were 2.21 and 3.82 (Fig. 3).

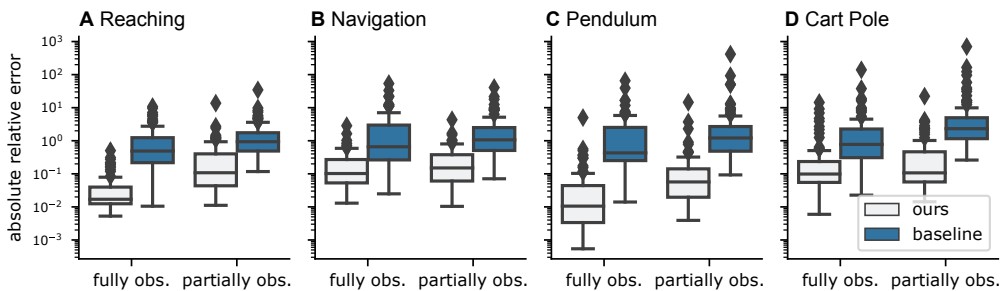

Figure 3: **Evaluation across tasks.** Absolute relative errors (log scale) for different tasks. Our method consistently outperforms the MCE baseline.

## 4.4 Information-seeking behavior in the light-dark domain

Finally, we investigate the ability of our method to disentangle sources of information-seeking behavior. In the light-dark domain [55], an agent moves in a 2D space and receives noisy measurements of its position, whose standard deviation depends on the horizontal distance from a light source:

$$\boldsymbol{y}_t = \boldsymbol{x}_t + \sigma |x_{t,1} - 5| \, \boldsymbol{w}_t, \tag{8}$$

where $\sigma$ governs the amount of perceptual uncertainty. This task is a common test for information-seeking behavior, because it requires the agent to move towards the light source and at the same time away from the target to reduce the uncertainty about its position relative to the target. When this uncertainty has been reduced, the agent can approach the target (see Appendix J.4 for details). The agent's goal is to reach the target's position **p** at the final time step, while minimizing control effort $\boldsymbol{u}_t^2$:

$$J = \underbrace{(\boldsymbol{x}_T - \mathbf{p})^2}_{\text{final cost}} + \underbrace{\sum_{t=1}^{T-1} \tfrac{1}{2}\boldsymbol{u}_t^2 + c\,(x_{t,1} - 5)^2}_{\text{running cost}}. \tag{9}$$

Different from the original problem formulation, we consider the case in which the agent may have an additional inherent desire to be close to the light, parameterized by $c$. We recover the original cost function [55] for $c = 0$, but we can represent agents that seek light more than necessary to reduce uncertainty for reaching the goal state with $c > 0$. Accordingly, both the reduction of perceptual uncertainty and state-dependent running cost could encourage the agent to move towards the light source before approaching the target. For an external observer, e.g., an experimenter, observing an agent moving towards the light might not reveal the different potential sources for this behavior. We now ask if it is possible to disentangle these two factors using our proposed IOC algorithm.

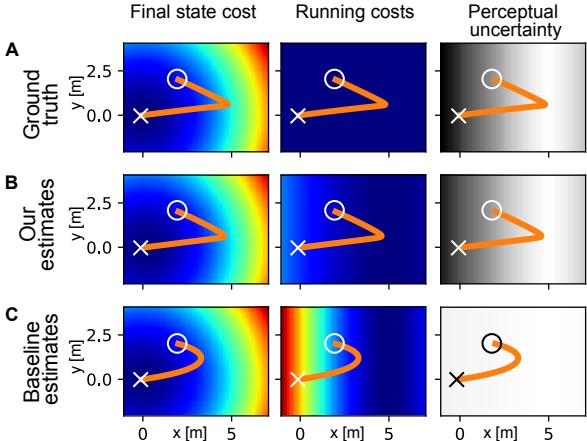

Figure 4: **Light-dark domain.** Cost maps and perceptual uncertainty map plotted with true parameters (**A**) and inferred parameters using our method (**B**) and baseline (**C**). Mean trajectories with start point (circle) and target (cross) are shown in orange.

We simulated 100 trajectories of an iLQG agent [partially observable version, 43] with no inherent desire to be near the light source ($c = 0$) and some perceptual uncertainty ($\sigma = 0.2$) depending on the distance to the light source. The agent first moves towards the light source and then reaches the target (Fig. 4 A). We inferred the parameters $\sigma$, **p**, and $c$ using our method and the baseline. Both methods recover the target position. Our method in addition infers values close to the true $c$, and $\sigma$. It therefore correctly attributes the information-seeking behavior of the agent to the perceptual uncertainty (Fig. 4 B). The baseline method, however, does not infer the correct

perceptual uncertainty. It instead attributes the agent's information-seeking behavior to an inherent desire to be in the right part of the room in the running cost (Fig. 4 C). This highlights the importance for IOC in partially observable domains to probabilistically take the agent's belief into account, if one is interested in inferring the correct cognitive mechanisms. For a quantitative evaluation of the maximum likelihood estimates in the light-dark domain, see Appendix K.4.

# 5   Conclusion

In this paper, we introduced a new IOC method for partially observable systems with stochastic non-linear dynamics and missing control signals. Using a probabilistic approach, we formulate the IOC problem as maximizing the likelihood of the observed states given the parameters. As the exact evaluation of the likelihood for a general non-linear model is intractable, we developed an efficient approximate likelihood by linearizing the system locally around the given trajectories, as in popular approaches such as the EKF or iLQG. By maintaining a distribution that tracks the agent's belief, an approximate likelihood can be evaluated in closed form within a single forward pass and efficiently optimized. Our proposed formulation is able to incorporate multiple sources of the stochasticity of the agent, reconciling the theory of past MCE IOC algorithms [e.g., 3] and approaches where the agent's stochasticity stems from an explicit stochastic observation and control model [29].

We evaluated our method on two stochastic variants of classic control tasks, pendulum and cart pole, and on two human behavioral tasks, a reaching and a navigation task. In comparison to an MCE baseline, we have found our method to achieve lower estimation errors across all tasks. Further, it successfully inferred noise parameters of the system, which was not possible with the baseline. In the light-dark domain, we were able to infer costs and perceptual uncertainty parameters, two different causes of apparent information-seeking behavior that could lead to qualitatively similar trajectories. This means that our method is a first step towards distinguishing pragmatic from epistemic controls. To further investigate this, it would be fruitful to examine belief-space planning methods that explicitly include the agent's belief covariance in the policy [56].

The limitations of our method are mainly due to the linearization of the dynamical system and the Gaussian approximations involved in the belief tracking formulation of the likelihood function. In more complex scenarios with non-Gaussian belief distributions, e.g., multimodal beliefs, the method will likely produce inaccurate results. This problem could be addressed by replacing the closed-form Gaussian belief by particle-based methods [57]. Further, we focused on tasks which could be solved well by control methods based on linearization and Gaussian approximation (iLQG and EKF), motivated by their popularity in applications in cognitive science and neuroscience. Forward problems that cannot be solved using iLQG are probably not directly solvable using our inverse method. While, in principle, our method is also applicable to other forward control methods that compute differentiable policies, it is an empirical question whether linearizing these policies leads to accurate approximate likelihoods and parameter estimates. A further limitation of our method is that it requires parametric models of the dynamics and noise structure. While missing parameters can be determined using our method, in the case of completely unknown dynamics a model-free approach to IOC would be more suitable. Lastly, while we have shown that inference is feasible, the results probably do not scale to high-dimensional parameter spaces. One reason for this is that optimization in a high-dimensional non-linear space can potentially get stuck in local minima. This problem could be relieved by using more advanced optimization methods. A further, more fundamental, concern with higher-dimensional parameter spaces is that identifiability issues and ambiguous solutions arise. However, our probabilistic approach with a closed-form likelihood opens up the possibility of using Bayesian methods to investigate the identifiability of model parameters [58]. For future work exploring the relationship to methods for learning world models in POMDPs might also be a fruitful direction [59, 60].

Our method provides a tool for researchers, e.g., in sensorimotor domains, to model sequential behavior by inferring an agent's subjective costs and internal uncertainties. This will enable answering novel scientific questions about how these quantities are affected by different experimental conditions, how they deviate from intended task goals and provided task instructions, or how they vary between individuals. This is particularly relevant to a computational understanding of naturalistic behavior [61–63], for which subjective utilities are mostly unknown.

## Acknowledgments and Disclosure of Funding

We gratefully acknowledge the computing time on the high-performance computer Lichtenberg at the NHR Centers NHR4CES at TU Darmstadt, and financial support by the project "Whitebox" funded by the Priority Program LOEWE of the Hessian Ministry of Higher Education, Science, Research and Art. We thank Fabian Kessler for helpful discussions and comments and an anonymous reviewer of a previous version of the manuscript for surmising that the present method could not infer information-seeking behavior.

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

# Appendix

## A  Notation

Table 1: Notation

| | |
|---|---|
| $\boldsymbol{x}_t \in \mathbb{R}^n$ | state at time $t$ |
| $\boldsymbol{u}_t \in \mathbb{R}^u$ | control at time $t$ |
| $\boldsymbol{y}_t \in \mathbb{R}^m$ | observation at time $t$ |
| $T \in \mathbb{N}$ | number of time steps |
| $f : \mathbb{R}^n \times \mathbb{R}^u \times \mathbb{R}^v \to \mathbb{R}^n$ | system dynamics function |
| $h : \mathbb{R}^n \times \mathbb{R}^w \to \mathbb{R}^m$ | observation function |
| $\boldsymbol{v}_t \in \mathbb{R}^v$ | standard multivariate normal dynamics noise |
| $\boldsymbol{w}_t \in \mathbb{R}^w$ | standard multivariate normal observation noise |
| $c_t : \mathbb{R}^n \times \mathbb{R}^u \to \mathbb{R}$ | cost function at intermediate time steps |
| $c_T : \mathbb{R}^n \to \mathbb{R}$ | cost function at final time step |
| $\mathbf{b}_t \in \mathbb{R}^b$ | summary statistics of agent's belief, $p(\boldsymbol{x}_t \mid \boldsymbol{y}_{1:t-1})$ |
| $\beta_t : \mathbb{R}^b \times \mathbb{R}^u \times \mathbb{R}^m \to \mathbb{R}^b$ | belief dynamics |
| $\pi_t : \mathbb{R}^b \times \mathbb{R}^j \to \mathbb{R}^u$ | policy of the agent |
| $\boldsymbol{\theta} \in \mathbb{R}^p$ | model parameters |
| $g : \mathbb{R}^n \times \mathbb{R}^b \times \mathbb{R}^v \times \mathbb{R}^w \times \mathbb{R}^j \to \mathbb{R}^n \times \mathbb{R}^b$ | joint dynamics of states and beliefs |

## B  MCE IRL Baseline

Maximum causal entropy inverse reinforcement learning (MCE IRL) [47] can be formulated as maximizing the likelihood

$$p(\boldsymbol{x}_{1:T} \mid \boldsymbol{\theta}) = p(\boldsymbol{x}_0 \mid \boldsymbol{\theta}) \prod_{t=0}^{T-1} p(\boldsymbol{x}_{t+1} \mid \boldsymbol{x}_t, \boldsymbol{u}_t, \boldsymbol{\theta}) \Pi_t^{\boldsymbol{\theta}}(\boldsymbol{u}_t \mid \boldsymbol{x}_t), \tag{10}$$

with

$$\Pi_t^{\boldsymbol{\theta}}(\boldsymbol{u}_t \mid \boldsymbol{x}_t) = \exp(Q_t^{\boldsymbol{\theta}}(\boldsymbol{x}_t, \boldsymbol{u}_t) - V_t^{\boldsymbol{\theta}}(\boldsymbol{x}_t)).$$

Here, $Q_t^{\boldsymbol{\theta}}$ is the soft Q-function at time $t$, given by

$$Q_t^{\boldsymbol{\theta}}(\boldsymbol{x}_t, \boldsymbol{u}_t) = -c_t(\boldsymbol{x}_t, \boldsymbol{u}_t) - \mathbb{E}[V_{t+1}^{\boldsymbol{\theta}}(\boldsymbol{x}_{t+1})]$$

and $V_t^{\boldsymbol{\theta}}$ the normalization,

$$V_t^{\boldsymbol{\theta}}(\boldsymbol{x}_t) = \log \int_{\boldsymbol{u}_t} \exp(Q_t^{\boldsymbol{\theta}}(\boldsymbol{x}_t, \boldsymbol{u}_t)) \, \mathrm{d}\boldsymbol{u}_t.$$

For arbitrary systems, computing the soft Q-function exactly is infeasible, therefore common methods apply approximations such as linearization [18], importance sampling [19], or deep function approximation [23]. For the case of linear dynamics and quadratic reward, the optimal policy is given by a Gaussian distribution $\Pi_t(\boldsymbol{u}_t \mid \boldsymbol{x}_t) = \mathcal{N}(\boldsymbol{u}_t; L_t\boldsymbol{x}_t, -H_t^{-1})$, where $L_t$ is the controller gain and $H_t$ a matrix resulting from the computation of the LQG controller [48]. As the tasks we consider can be well solved by linearizing the dynamics locally, we choose an approximation by linearization and use the optimal MCE controller for the linearized dynamics to compute the likelihood function for the parameter set $\boldsymbol{\theta}$.

To apply this baseline to the setting where control signals are missing, we use the estimates of the controls as we determine in our proposed method for the data-based linearization (Section 3.3 and Appendix G).

We can compute the approximate likelihood function by performing the following steps:

1. Estimate the missing control signals using Eq. (11)

2. Linearize the system as described in Section 3.3
3. Compute the MCE policy for the linearized system, i.e., compute $Q_t^{\boldsymbol{\theta}}$ and $V_t^{\boldsymbol{\theta}}$
4. Compute the likelihood (in log space) using Eq. (10)

To maximize the (log) likelihood efficiently, one needs to compute the gradient of the likelihood function, which is straightforwardly achieved by backpropagating the gradient using automatic differentiation in step 4.

Note that there is no explicit model of partial observability and we use point estimates for the control signals, therefore the likelihood in Eq. (10) decomposes in independent factors given states and controls. In contrast, in our approach, when incorporating partial observability and missing controls, one has to compute approximate likelihood contributions within a forward pass.

## C   Algorithm to compute an approximate likelihood for IOC

---

**Algorithm 1** Approximate likelihood computation

---

**Output:** Approximate likelihood of parameters $p(\boldsymbol{x}_{1:T} \mid \boldsymbol{\theta})$
**Input:** Parameters $\boldsymbol{\theta}$, Data $\boldsymbol{x}_{1:T}$, Model $f, h$
 1: Determine the policy $\pi$ using iLQG
 2: Determine the belief dynamics $\beta$ using the EKF
 3: **for** $t$ in $\{1, \ldots, T-1\}$ **do**
 4:     Compute $p(\boldsymbol{x}_{t+1}, \mathbf{b}_{t+1} \mid \boldsymbol{x}_{1:t})$ using Eq. (4)
 5:     Update $p(\mathbf{b}_{t+1} \mid \boldsymbol{x}_{1:t+1})$ using Eq. (5)
 6:     Obtain $p(\boldsymbol{x}_{t+1} \mid \boldsymbol{x}_{1:t})$ using Eq. (3)
 7: **end for**

---

## D   Inverse optimal control derivations

We start with defining the joint dynamics of states and estimates, which returns the next state and estimate as a function of the previous state and estimate and the noises,

$$\begin{bmatrix} \boldsymbol{x}_{t+1} \\ \mathbf{b}_{t+1} \end{bmatrix} = g(\boldsymbol{x}_t, \mathbf{b}_t, \boldsymbol{v}_t, \boldsymbol{w}_t, \boldsymbol{\xi}_t).$$

To do so, we insert the policy and observation function into the system and belief dynamics, giving

$$\boldsymbol{x}_{t+1} = f(\boldsymbol{x}_t, \pi_t(\mathbf{b}_t, \boldsymbol{\xi}_t), \boldsymbol{v}_t),$$
$$\mathbf{b}_{t+1} = \beta_t(\mathbf{b}_t, \pi_t(\mathbf{b}_t, \boldsymbol{\xi}_t), h(\boldsymbol{x}_t, \boldsymbol{w}_t)).$$

The individual factors of the likelihood function can be determined as

$$p(\boldsymbol{x}_{t+1} \mid \boldsymbol{x}_{1:t}) = \int p(\boldsymbol{x}_{t+1}, \mathbf{b}_{t+1} \mid \boldsymbol{x}_{1:t}) \, \mathrm{d}\mathbf{b}_{t+1},$$

where $p(\boldsymbol{x}_{t+1}, \mathbf{b}_{t+1} \mid \boldsymbol{x}_{1:t})$ is given by marginalizing over the current belief $\mathbf{b}_t$ as

$$p(\boldsymbol{x}_{t+1}, \mathbf{b}_{t+1} \mid \boldsymbol{x}_{1:t}) = \int p(\boldsymbol{x}_{t+1}, \mathbf{b}_{t+1}, \mathbf{b}_t \mid \boldsymbol{x}_{1:t}) \, \mathrm{d}\mathbf{b}_t$$
$$= \int p(\boldsymbol{x}_{t+1}, \mathbf{b}_{t+1} \mid \boldsymbol{x}_{1:t}, \mathbf{b}_t) \, p(\mathbf{b}_t \mid \boldsymbol{x}_{1:t}) \, \mathrm{d}\mathbf{b}_t$$
$$= \int p(\boldsymbol{x}_{t+1}, \mathbf{b}_{t+1} \mid \boldsymbol{x}_t, \mathbf{b}_t) \, p(\mathbf{b}_t \mid \boldsymbol{x}_{1:t}) \, \mathrm{d}\mathbf{b}_t.$$

**Linearization**

To derive a tractable approximation of the distribution $p(\boldsymbol{x}_{t+1}, \mathbf{b}_{t+1} \mid \boldsymbol{x}_{1:t})$, we model the initial belief of the agent's state estimate $p(\mathbf{b}_1)$ as a Gaussian and linearize the joint dynamics function, leading to a Gaussian approximation of the desired quantity, i.e., $p(\boldsymbol{x}_{t+1}, \mathbf{b}_{t+1} \mid \boldsymbol{x}_{1:t}) \approx \mathcal{N}(\mu_t, \Sigma_t)$.

First, we apply a first-order Taylor expansion of the joint dynamics $g$ around the observed state $\boldsymbol{x}_t$ and the mean of the belief $\mu_t^{(b)}$ and the noises:

$$g(\boldsymbol{x}_t, \mathbf{b}_t, \boldsymbol{v}_t, \boldsymbol{w}_t, \boldsymbol{\xi}_t) \approx g(\boldsymbol{x}_t, \mu_t^{(b)}, 0, 0, 0) + J_{\mathbf{b}}(\mathbf{b}_t - \mu_t^{(b)}) + J_{\boldsymbol{v}}\boldsymbol{v}_t + J_{\boldsymbol{w}}\boldsymbol{w}_t + J_{\boldsymbol{\xi}}\boldsymbol{\xi}_t,$$

where $J_{\bullet}$ denotes the Jacobian of $g$ w.r.t. $\bullet$, evaluated at $(\boldsymbol{x}_t, \mu_t^{(b)}, 0, 0, 0)$.

To derive an explicit representation of the Jacobians, we insert the filtering and control law obtained by the Kalman filter and maximum entropy iLQG controller and find

$$g(\boldsymbol{x}_t, \mathbf{b}_t, \boldsymbol{v}_t, \boldsymbol{w}_t, \boldsymbol{\xi}_t) = \begin{bmatrix} f(\boldsymbol{x}_t, \pi_t(\mathbf{b}_t, \boldsymbol{\xi}_t), \boldsymbol{v}_t) \\ \beta_t(\mathbf{b}_t, \pi_t(\mathbf{b}_t, \boldsymbol{\xi}_t), h(\boldsymbol{x}_t, \boldsymbol{w}_t)) \end{bmatrix}$$

$$= \begin{bmatrix} f(\boldsymbol{x}_t, \boldsymbol{u}_t, \boldsymbol{v}_t) \\ \beta_t(\mathbf{b}_t, \boldsymbol{u}_t, \boldsymbol{y}_t) \end{bmatrix},$$

with

$$\boldsymbol{y}_t = h(\boldsymbol{x}_t, \boldsymbol{w}_t),$$

$$\boldsymbol{u}_t = \pi_t(\mathbf{b}_t, \boldsymbol{\xi}_t) = L_t(\mathbf{b}_t - \bar{\boldsymbol{x}}_{1:T}) + \mathbf{m}_t + \bar{\boldsymbol{u}}_{1:T} - \tilde{L}_t\boldsymbol{\xi}_t,$$

$$\beta_t(\mathbf{b}_t, \boldsymbol{u}_t, \boldsymbol{y}_t) = f(\mathbf{b}_t, \boldsymbol{u}_t, 0) + K_t(\boldsymbol{y}_t - h(\mathbf{b}_t, 0)),$$

leading to the equations

$$\mathbb{J}_{\mathbf{b}} = \begin{bmatrix} \nabla_{\boldsymbol{u}}f(\boldsymbol{x}_t, \boldsymbol{u}_t, 0)\nabla_{\mathbf{b}}\boldsymbol{u}_t \\ \nabla_{\mathbf{b}}\beta_t(\mathbf{b}_t, \boldsymbol{u}_t, \boldsymbol{y}_t) + \nabla_{\boldsymbol{u}}\beta_t(\mathbf{b}_t, \boldsymbol{u}_t, \boldsymbol{y}_t)\nabla_{\mathbf{b}}\boldsymbol{u}_t \end{bmatrix}$$

$$= \begin{bmatrix} \nabla_{\boldsymbol{u}}f(\boldsymbol{x}_t, \boldsymbol{u}_t, 0)L_t \\ \nabla_{\boldsymbol{x}}f(\mathbf{b}_t, \boldsymbol{u}_t, 0) - K_t\nabla_{\boldsymbol{x}}h(\mathbf{b}_t, 0) + \nabla_{\boldsymbol{u}}f(\mathbf{b}_t, \boldsymbol{u}_t, 0)L_t \end{bmatrix},$$

$$\mathbb{J}_{\boldsymbol{v}} = \begin{bmatrix} \nabla_{\boldsymbol{v}}f(\boldsymbol{x}_t, \boldsymbol{u}_t, 0) \\ \nabla_{\boldsymbol{v}}f(\mathbf{b}_t, \boldsymbol{u}_t, 0) \end{bmatrix},$$

$$\mathbb{J}_{\boldsymbol{w}} = \begin{bmatrix} 0 \\ \nabla_h\beta_t(\mathbf{b}_t, \boldsymbol{u}_t, \boldsymbol{y}_t)\nabla_{\boldsymbol{w}}h(\mathbf{b}_t, 0) \end{bmatrix} = \begin{bmatrix} 0 \\ -K_t\nabla_{\boldsymbol{w}}h(\mathbf{b}_t, 0) \end{bmatrix},$$

$$\mathbb{J}_{\boldsymbol{\xi}} = \begin{bmatrix} \nabla_{\boldsymbol{u}}f(\boldsymbol{x}_t, \boldsymbol{u}_t, 0)\nabla_{\boldsymbol{\xi}}\boldsymbol{u}_t \\ \nabla_{\boldsymbol{u}}f(\mathbf{b}_t, \boldsymbol{u}_t, 0)\nabla_{\boldsymbol{\xi}}\boldsymbol{u}_t \end{bmatrix} = \begin{bmatrix} \nabla_{\boldsymbol{u}}f(\boldsymbol{x}_t, \boldsymbol{u}_t, 0)L_t \\ \nabla_{\boldsymbol{u}}f(\mathbf{b}_t, \boldsymbol{u}_t, 0)L_t \end{bmatrix}.$$

Propagating the Gaussian belief over the agent's state estimate, $p(\mathbf{b}_t \mid \boldsymbol{x}_{1:t})$ through the linearized dynamics model (Eq. (4)) can be done by applying standard identities for linear transformations of Gaussian random variables [64] and gives

$$p(\boldsymbol{x}_{t+1}, \mathbf{b}_{t+1} \mid \boldsymbol{x}_{1:t}) \approx \mathcal{N}(\mu_t, \Sigma_t),$$

with $\mu_t = g(\boldsymbol{x}_t, \mu_t^{(b)}, 0, 0, 0)$ and $\Sigma_t = \mathbb{J}_{\mathbf{b}}\Sigma_t^{(b)}\mathbb{J}_{\mathbf{b}}^T + \mathbb{J}_{\boldsymbol{v}}\mathbb{J}_{\boldsymbol{v}}^T + \mathbb{J}_{\boldsymbol{w}}\mathbb{J}_{\boldsymbol{w}}^T + \mathbb{J}_{\boldsymbol{\xi}}\mathbb{J}_{\boldsymbol{\xi}}^T$. Marginalization over $\mathbf{b}_{t+1}$ gives the desired likelihood factor, while conditioning on $\boldsymbol{x}_{t+1}$ gives the belief statistic for the following time step (see Section 3.1).

# E  Special case: full observability

If the state $\boldsymbol{x}_t$ is fully observable to the agent, the problem is simplified significantly. The control problem from the agent's perspective (Fig. S1, left) can be solved by applying iLQG [8] to the observed states directly (Section 2.2)

$$\boldsymbol{u}_t = \pi_t(\boldsymbol{x}_t, \boldsymbol{\xi}_t),$$

This also simplifies the IOC problem from the researcher's perspective because evaluating the approximate likelihood becomes straightforward as we do not need to marginalize over the agent's belief.

Recall that the likelihood can be decomposed as

$$p(\boldsymbol{x}_{1:T} \mid \theta) = p(\boldsymbol{x}_0 \mid \theta) \prod_{t=0}^{T-1} p(\boldsymbol{x}_{t+1} \mid \boldsymbol{x}_{1:t}, \theta)$$

and the dynamics are given as

$$\boldsymbol{x}_{t+1} = f(\boldsymbol{x}_t, \boldsymbol{u}_t, \boldsymbol{v}_t) = f(\boldsymbol{x}_t, \pi(\boldsymbol{x}_t, \boldsymbol{\xi}_t), \boldsymbol{v}_t).$$

We can approximate the likelihood contribution at each time step as

$$p(\boldsymbol{x}_{t+1} \mid \boldsymbol{x}_{1:t}, \theta) \approx \mathcal{N}(\mu_t, \Sigma_t),$$

with

$$\mu_t = f(\boldsymbol{x}_t, \pi(\boldsymbol{x}_t, 0), 0)$$
$$\Sigma_t = \mathbb{J}_{\boldsymbol{v}} \mathbb{J}_{\boldsymbol{v}}^T + \mathbb{J}_{\boldsymbol{\xi}} \mathbb{J}_{\boldsymbol{\xi}}^T,$$

where $\mathbb{J}_{\bullet}$ denotes the Jacobian of $f$ w.r.t. $\bullet$, evaluated at $(\boldsymbol{x}_t, \pi(\boldsymbol{x}_t, 0), 0)$:

$$\mathbb{J}_{\boldsymbol{v}} = \nabla_{\boldsymbol{v}} f(\boldsymbol{x}_t, \boldsymbol{u}_t, 0),$$
$$\mathbb{J}_{\boldsymbol{\xi}} = \nabla_{\boldsymbol{u}} f(\boldsymbol{x}_t, \boldsymbol{u}_t, 0) \nabla_{\boldsymbol{\xi}} \boldsymbol{u}_t = \nabla_{\boldsymbol{u}} f(\boldsymbol{x}_t, \boldsymbol{u}_t, 0) L_t.$$

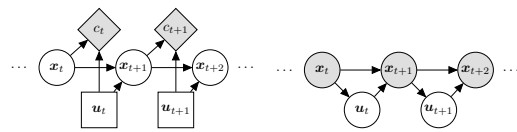

Figure S1: **Left:** decision network from the agent's perspective [following the notation used in 9]. The agent directly observes the state $\boldsymbol{x}_t$. At each time step, they perform a control $\boldsymbol{u}_t$ and incur a cost $c_t$. **Right:** probabilistic graphical model from the researcher's perspective, who observes a trajectory $\boldsymbol{x}_{1:T}$ from an agent. The control signals $\boldsymbol{u}_t$ are not directly observed.

# F  Implementation

We provide a flexible framework for defining non-linear stochastic dynamical systems of the form introduced in Section 2.1 by implementing the dynamics, observation function, and cost function. The optimal estimation control methods based on iterative linearization are implemented using the automatic differentiation library `jax` [65], so that Jacobians and Hessians for the linearization of the dynamics and quadratization of the cost do not have to be specified manually. Furthermore, the Jacobians needed for linear-Gaussian approximation in the IOC likelihood (Section 3.2) are also computed using automatic differentiation. Gradient-based maximization of the log likelihood was performed using the L-BFGS-B `scipy` [66] from the `jaxopt` library [67]. The implementation is provided in the supplementary material and will be made public upon publication.

For computations, we used an Intel Xeon Platinum 9242 Processor, using 1 core per run. The mean run times for computing maximum likelihood estimates (partially observable setting) were as follows:

- Reaching: 140 s
- Navigation: 64 s
- Pendulum: 31 s
- CartPole: 115 s

## G  Estimating controls for linearization

For every evaluation of the likelihood function as described in Section 3.1, we would naively need to solve the forward control problem once by running the iLQG algorithm given the current parameter values. This is computationally inefficient due to the iterative procedure of iLQG, which requires multiple forward and backward passes. Instead of performing the iterative procedure, which computes the locally optimal control law $\{L_{1:T-1}, \mathbf{m}_{1:T-1}\}$ around a nominal trajectory $\{\bar{\boldsymbol{x}}_{1:T}, \bar{\boldsymbol{u}}_{1_T-1}\}$, then computes the new nominal trajectory given the current control law and so on, we realize that in the IOC setting, we already have observed the actual trajectory $\boldsymbol{x}_{1:T}$ performed by the agent. To make solving the forward control problem more efficient, we only compute the locally optimal control law once around this trajectory. This would require the control signals $\boldsymbol{u}_{1:T}$ to be given, but they are unobserved in our problem setting. To obtain a proxy for the controls for the purposes of linearization, we solve for the controls given the trajectory $\boldsymbol{x}_{1:T}$ in the system dynamics equation $\boldsymbol{x}_{t+1} = f(\boldsymbol{x}_t, \boldsymbol{u}_t, 0)$ using the Gauss-Newton method for non-linear least squares as implemented in `jaxopt`, i.e.,

$$\hat{\boldsymbol{u}}_t = \arg\max_{\boldsymbol{u}} \left(\boldsymbol{x}_{t+1} - f\left(\boldsymbol{x}_t, \boldsymbol{u}, 0\right)\right)^2. \tag{11}$$

## H  Efficient gradient computation

In the case where the full state is not observable, one cannot linearize around the given trajectory as descibed in Section 3.3. The forward optimal control problem is commonly solved by starting with a randomly initialized nominal trajectory and iterating between computing the locally optimal control law and linearization until convergence. To compute gradients, a naive approach would be to differentiate through the loop of the forward procedure. In our experiments, we found, however, that this approach is numerically unstable and does not yield correct gradients (see Fig. S2). Here, we provide two approaches to compute gradients in this case.

### H.1  Differentiating through the last iteration

In this approach, one first solves the forward problem as usual to determine the linearization. One then executes one additional backward pass for the optimal controller, which one can differentiate using automatic differentiation.

### H.2  Applying the implicit function theorem

Determining the optimal linearization and controller is essentially a fixed point problem. To differentiate through the optimal linearization and controller, we apply the implicit function theorem [68, 69].

We assume the forward equation to determine the linearization is given by

$$\boldsymbol{x}_{t+1} = \hat{f}(\boldsymbol{x}_t, \boldsymbol{u}_t, \boldsymbol{\theta}),$$
$$\boldsymbol{u}_t = \hat{\pi}_t(\boldsymbol{x}_t, \boldsymbol{s}_t, \boldsymbol{\theta})$$

where $\boldsymbol{\theta}$ are the parameters we want to differentiate with respect to and $\boldsymbol{s}_t$ is a vector determining the optimal controller, such that it can be determined via a backward pass with

$$\boldsymbol{s}_{t-1} = \gamma(\boldsymbol{x}_t, \boldsymbol{u}_t, \boldsymbol{s}_t, \boldsymbol{\theta}).$$

For the LQG controller, $\boldsymbol{s}_t$ would consist of the (vectorized) matrix characterizing the optimal value function.

The fixed point problem of jointly determining the optimal linearization and controller can be formulated as determining $X := [\boldsymbol{u}_1, \boldsymbol{x}_2, \boldsymbol{u}_2, \ldots, \boldsymbol{u}_{T-1}, \boldsymbol{x}_T, \boldsymbol{s}_1, \ldots, \boldsymbol{s}_T]$ such that

$$X = \eta(X, \boldsymbol{\theta}) := [\hat{\pi}_1(\boldsymbol{x}_1, \boldsymbol{s}_1, \boldsymbol{\theta}), \hat{f}(\boldsymbol{x}_1, \boldsymbol{u}_1, \boldsymbol{\theta}), \hat{\pi}_2(\boldsymbol{x}_2, \boldsymbol{s}_2, \boldsymbol{\theta}), \ldots, \hat{f}(\boldsymbol{x}_{T-1}, \boldsymbol{u}_{T-1}, \boldsymbol{\theta}),$$
$$\gamma(\boldsymbol{x}_2, \boldsymbol{u}_2, \boldsymbol{s}_2, \boldsymbol{\theta}), \ldots, \gamma(\boldsymbol{x}_T, \boldsymbol{u}_T, \boldsymbol{s}_T, \boldsymbol{\theta})].$$

The optimal solution of the fixed point problem satisfies

$$\kappa(X^*(\boldsymbol{\theta}), \boldsymbol{\theta}) := X^*(\boldsymbol{\theta}) - \eta(X^*(\boldsymbol{\theta}), \boldsymbol{\theta}) = 0,$$

where $X^*(\boldsymbol{\theta})$ is a function yielding the optimal solution depending on $\boldsymbol{\theta}$, i.e., $X^* : \mathbb{R}^p \to \mathbb{R}^{(n+u+s)t}$, where $s$ is the number of elements of $\boldsymbol{s}_t$.

The implicit function theorem [68] gives, that for $(X_0, \boldsymbol{\theta}_0)$ satisfying $\kappa(X_0, \boldsymbol{\theta}_0) = 0$ with the requirement that $\kappa$ is continuously differentiable and the Jacobian $\partial_{\boldsymbol{\theta}} \kappa(X_0, \boldsymbol{\theta}_0)$ is a square invertible matrix, then there exists a function $X^*(\cdot)$ on a neighborhood of $\boldsymbol{\theta}_0$ such that $X^*(\boldsymbol{\theta}_0) = X_0$. Furthermore, for all $\boldsymbol{\theta}$ in this neighborhood, we have that $\kappa(X^*(\boldsymbol{\theta}), \boldsymbol{\theta}) = 0$ and $\partial X^*(\boldsymbol{\theta})$ exists.

By applying the chain rule, the Jacobian $\partial X^*(\boldsymbol{\theta})$ needs to satisfy

$$
\begin{aligned}
0 &= \partial_1 \kappa(X^*(\boldsymbol{\theta}), \boldsymbol{\theta}) \partial X^*(\boldsymbol{\theta}) + \partial_2 \kappa(X^*(\boldsymbol{\theta}), \boldsymbol{\theta}) \\
&= (I - \partial_1 \eta(X^*(\boldsymbol{\theta}), \boldsymbol{\theta})) \partial X^*(\boldsymbol{\theta}) - \partial_2 \eta(X^*(\boldsymbol{\theta}), \boldsymbol{\theta}).
\end{aligned}
$$

$\partial_1 \eta(X^*(\boldsymbol{\theta}), \boldsymbol{\theta}))$ is given by a sparse matrix, as each element of $\eta(X, \boldsymbol{\theta})$ only depends on few elements of $X$. The desired gradient $\partial X^*(\boldsymbol{\theta})$ (or vector-Jacobian products with it) can then be computed using linear system solvers [69].

### H.3 Results with different gradient computation methods

We ran the MLE for the partially observable reaching task with different gradient computation methods. The results are shown in Fig. S2. While unrolling the loop led to numerical problems, so the estimates are far off the true values, all other methods led to quite similar results. The mean run times were as follows:

- Unrolled loop: 141 s
- Differentiating through the last iteration: 103 s
- Implicit differentiation: 243 s
- Data-based linearization: 129 s

## I Hyperparameters

Throughout the experiments, we have used the following hyperparameters:

Table 2: Hyperparameters

| | |
|---|---|
| Data set size (number of trajectories) | 50 |
| Number of datasets per evaluation | 100 |
| Number of time steps ($T$) | 50 (reaching, navigation, pendulum, light-dark), 200 (cartpole) |
| Optimizer | L-BFGS-B (`scipy` wrapper from `jaxopt`) |
| Maximum entropy temperature | $10^{-6}$ (reaching, navigation), $10^{-3}$ (pendulum, cartpole), $10^{-5}$ (light-dark) |
| Optimizer restarts | 50 |

## J Tasks

### J.1 Reaching task with biomechanical arm model

We implemented the non-linear biomechanical model for arm movements from Todorov and Li [8] and its partially observed version from Li and Todorov [43], which are described in more detail in the PhD thesis of Li [70]. The dynamics describe the movement of a two-link arm, which can be controlled by applying torques to the two joints. The task is to move the hand to a target location, as defined in the cost function

$$
J = \|\mathbf{e}_T - \mathbf{e}^*\|^2 + c_v \|\dot{\mathbf{e}}_T\|^2 + c_a \sum_{t=1}^{T-1} \|\mathbf{u}_t\|^2 ,
$$

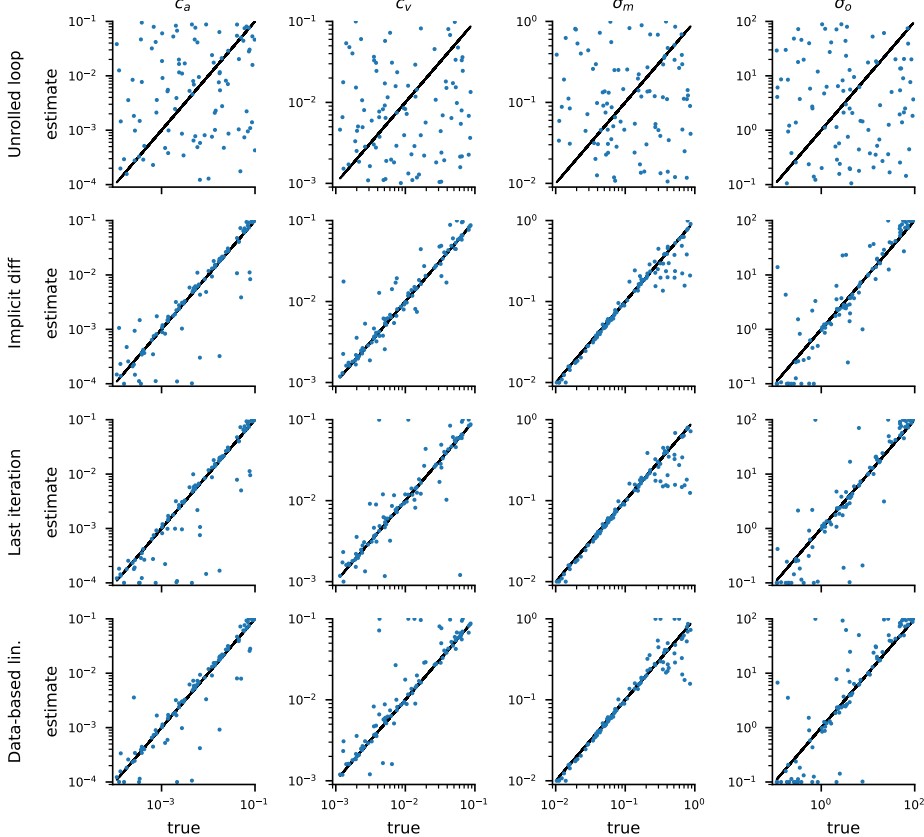

Figure S2: Maximum likelihood parameter estimates for the partially observable reaching task with different gradient computation methods

where $\mathbf{e}_T$ is the position of the hand at the final time step in Cartesian coordinates, $\mathbf{e}^\star$ is the position of the target, and $\dot{\mathbf{e}}_T$ is the final velocity of the hand. That is, the task is to bring the hand to the target at the final time step ($T = 50$) and minimize the final velocity while making as little control as possible along the way. The parameters $c_v$ and $c_a$ trade off, how important the velocity and control parts of the cost function are relative to the main goal of reaching the target. Additionally, we parameterize the signal-dependent variability of the motor system with the parameter $\sigma_m$, which defines how strongly the variability is scaled by the control magnitude.

## J.2  Navigation task

We implemented a simple navigation task in which an agent walks towards a target. The state $\boldsymbol{x}_t = [x_t, y_t, \theta_t, \omega_t]^T$ consists of the position in the horizontal plane, the heading angle, and the speed in the heading direction. The dynamics are

$$f(\boldsymbol{x}, \boldsymbol{u}, \boldsymbol{v}) = \boldsymbol{x} + dt \begin{bmatrix} \cos(\theta) \cdot \omega \\ \sin(\theta) \cdot \omega \\ u_1 + \sigma_m u_1 v_1 \\ u_2 + \sigma_m u_2 v_2 \end{bmatrix},$$

which means that the agent controls the velocity of the heading angle and the acceleration in heading direction. There is control-dependent noise, whose strength is determined by the parameter $\sigma_m$. The observation model is

$$h(\boldsymbol{x}, \boldsymbol{w}) = \begin{bmatrix} \sqrt{(x - k_1)^2 + (y - k_2)^2} \\ \tan^{-1}(y - k_2, x - k_1) \\ \omega \end{bmatrix} + \sigma_o \boldsymbol{w},$$

where $\mathbf{k} = [k_1, k_2]^T$ is the position of the target and $\sigma_o$ determines the magnitude of the observation noise. The cost function is

$$J = (x_T - k_1)^2 + (y_T - k_2)^2 + c_v \omega_T + \sum_{t=1}^{T-1} c_a \boldsymbol{u}_t^T \boldsymbol{u}_t,$$

with parameters determining the cost of controls ($c_a$) and the cost of the final velocity ($c_v$). The time horizon was $T = 50$.

## J.3 Classic control tasks

We evaluate our method on two classic continuous control tasks. Specifically, we build upon the Pendulum and Cart Pole environments from the gym library [54]. Because these environments are not stochastic in their standard implementations and are typically used in an infinite-horizon reinforcement learning setting, we made the following changes to the environments. To account for the finite-horizon setting we are considering in this work, the state-dependent costs associated with the task goal are applied only to the final time step, while control-dependent costs are applied at every time step along the trajectory. To make the problems stochastic, we have added noise on the dynamics and the observation function.

### J.3.1 Pendulum

The task is to make a pendulum, which is attached to a fixed point on one side, swing into an upright position, i.e. to reach an angle $\theta = 0$. The pendulum starts at the bottom ($\theta = \pi$) and can be controlled by applying a torque to the free end of the pendulum. We added control-dependent noise to the dynamics, where the parameter $\sigma_m$ controls, how strongly a standard normal noise is scaled by the magnitude of the torque. In addition to this motor noise parameter, the task has two other free parameters: the cost of controls $c_a$ and the cost of the final velocity $c_v$. In the partially observable version of the task, the agent receives the observation via the non-linear stochastic observation model $\mathbf{y}_t = \left[\sin \theta_t, \cos \theta_t, \dot{\theta}_t\right]^T + 0.1 \boldsymbol{w}_t$. The time horizon was $T = 50$.

### J.3.2 Cart pole

In the Cart Pole task, a pole is attached to a cart that can be moved left or right by applying a continuous force. In our version of the task, the goal is to move the cart from the horizontal position $x = 0$ to $x = 1$ while balancing the pole in an upright position. Again, we add control-dependent noise to the force, parameterized the strength of the linear control-dependence $\sigma_m$. As above, the other two parameters are the control cost $c_a$ and the cost of the final velocity $c_v$. In the partially observed version of the task, the agent receives a noisy observation with $\mathbf{y}_t = \mathbf{x}_t + \boldsymbol{w}_t$. The time horizon was $T = 200$.

## J.4 Light-dark domain

We use a slightly modified version of the light-dark domain, which was originally introduced by Platt et al. [55]. We adapted the light-dark domain's motion model to include signal-dependent noise on the control

$$\boldsymbol{x}_{t+1} = \boldsymbol{x}_t + dt\,\boldsymbol{u}_t + 0.1\,\boldsymbol{u}_t \odot \boldsymbol{v}_t$$

The standard deviation of the perceptual uncertainty varies with the horizontal distance to the light source:

$$\boldsymbol{y}_t = \boldsymbol{x}_t + \sigma|x_{t,1} - 5|\,\boldsymbol{w}_t.$$

The cost function is composed of three terms. First, the agent should minimize the squared distance to the target $\mathbf{p}$ at the final time step $T$. Second, the agent should minimize the squared control signals $\boldsymbol{u}_t$. And finally, the agent should minimize the horizontal distance to the light source:

$$J = \underbrace{(\boldsymbol{x}_T - \mathbf{p})^2}_{\text{final cost}} + \underbrace{\sum_{t=1}^{T-1} \tfrac{1}{2}\boldsymbol{u}_t^2 + c\,(x_{t,1} - 5)^2}_{\text{running cost}}.$$

The time-horizon was set to $T = 50$.

We simulated the behavior of two agents in the light-dark domain in Fig. S3. First, the partially observable version of iLQG [43] takes the expected belief covariance into account for the computation of the policy. This agent moves towards the light source before approaching the target. Adding another cost term $c$ that expresses a preference to be close to the light source does not change much about this behavior (top row). Second, the fully observable version of iLQG [8] computes the policy irrespective of the belief. To apply it in the partially observable setting, we simply combine it with an EKF for the state estimation. This agent does not move towards the light source by default and as a result, the belief uncertainty is higher. The agent only moves towards the light source before approaching the target if we add the additional cost term $c > 0$. One can also see in Fig. S3 that the agent's belief is more uncertain if it does not move to the light source before approaching the target. This results in a higher variability around the final position (inset plots in Fig. S3).

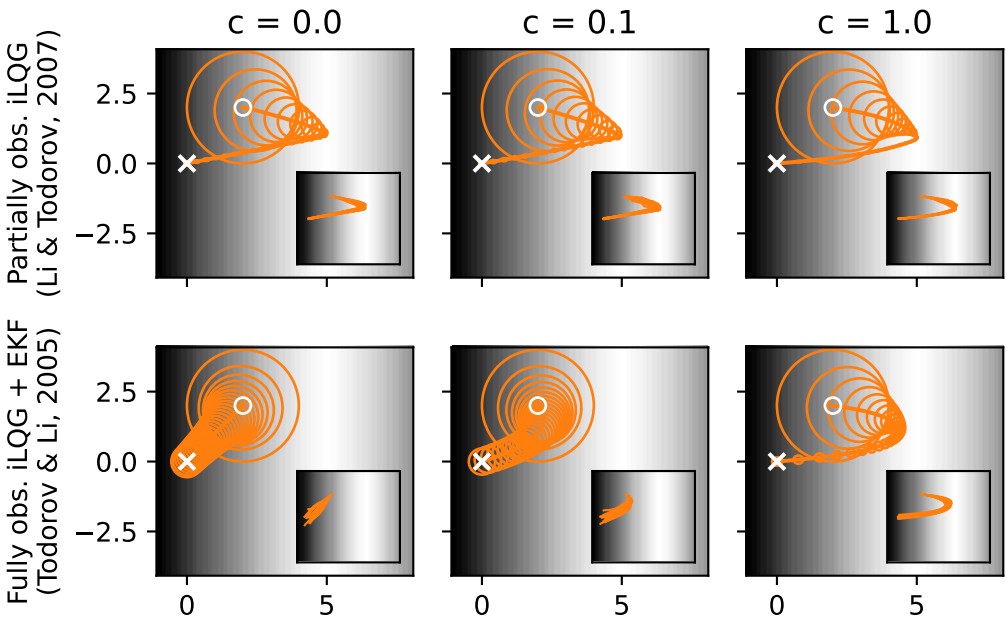

Figure S3: **Trajectories and beliefs in the light-dark domain.** Partially observable (top) vs. fully observable (bottom) iLQG in the light dark domain with different values for the cost parameter $c$, which expresses an inherent desire to be near the light source. The circles indicate the agent's belief covariance (2 standard deviations). One can also see that the variability around the final position is higher for agents that do not move towards the light source before approaching the target (inset plots).

# K Additional results

The parameter estimates and true values are provided in Fig. S4, Fig. S5, Fig. S6.

## K.1 Results for all tasks in the partially observable setting

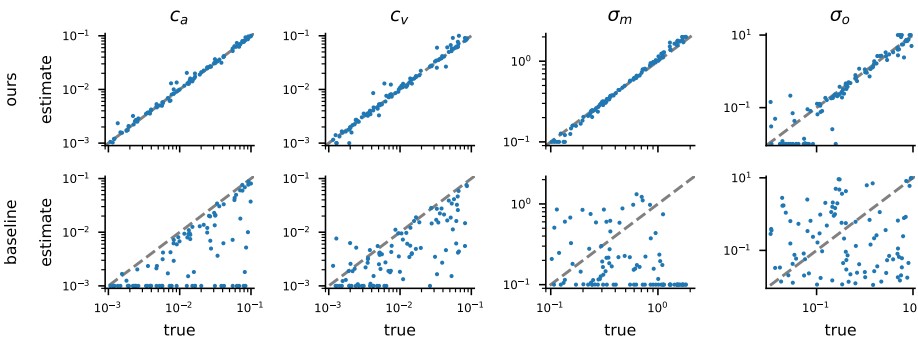

Figure S4: Maximum likelihood parameter estimates for the Pendulum task (partially observable cases).

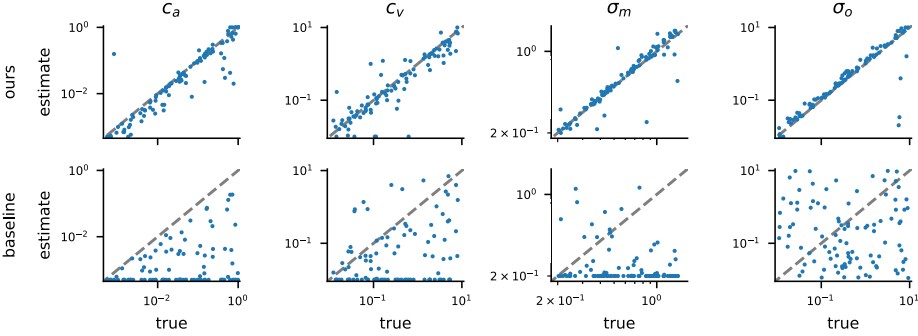

Figure S5: Maximum likelihood parameter estimates for the Cart Pole task (partially observable).

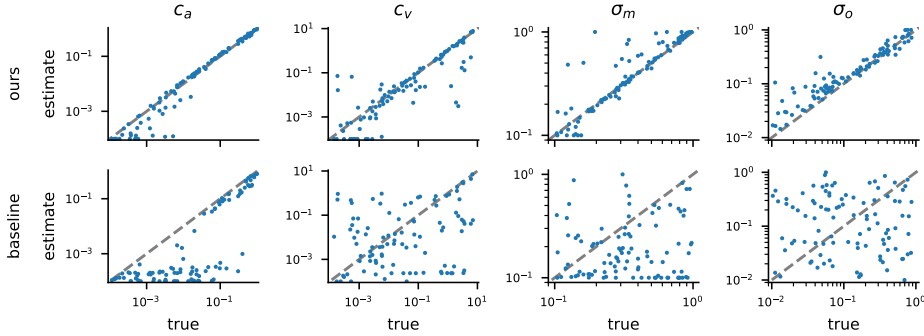

Figure S6: Maximum likelihood parameter estimates for the navigation task (partially observable).

## K.2 Results for all tasks in the fully observable setting

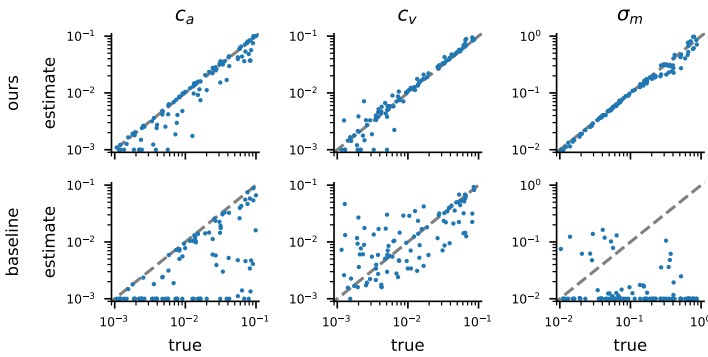

Figure S7: Maximum likelihood parameter estimates for the reaching task (fully observable).

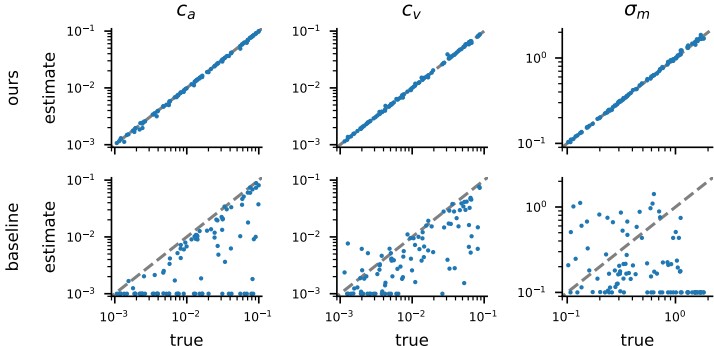

Figure S8: Maximum likelihood parameter estimates for the pendulum task (fully observable).

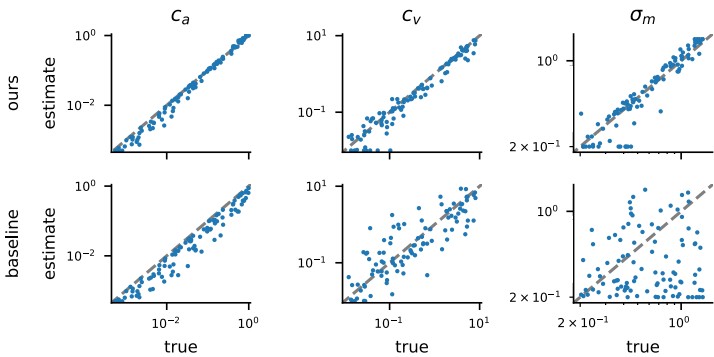

Figure S9: Maximum likelihood parameter estimates for the cart pole task (fully observable).

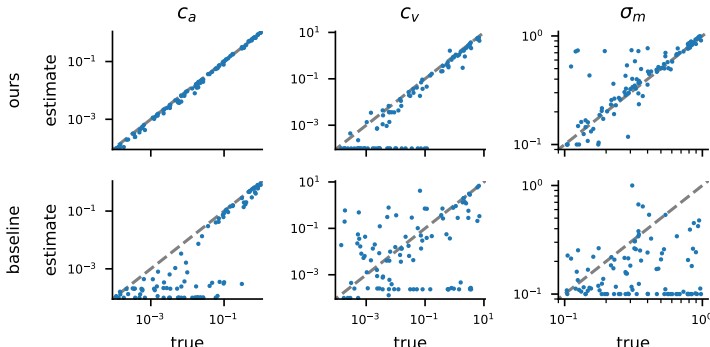

Figure S10: Maximum likelihood parameter estimates for the navigation task (fully observable).

## K.3 Results for baseline with given control signals

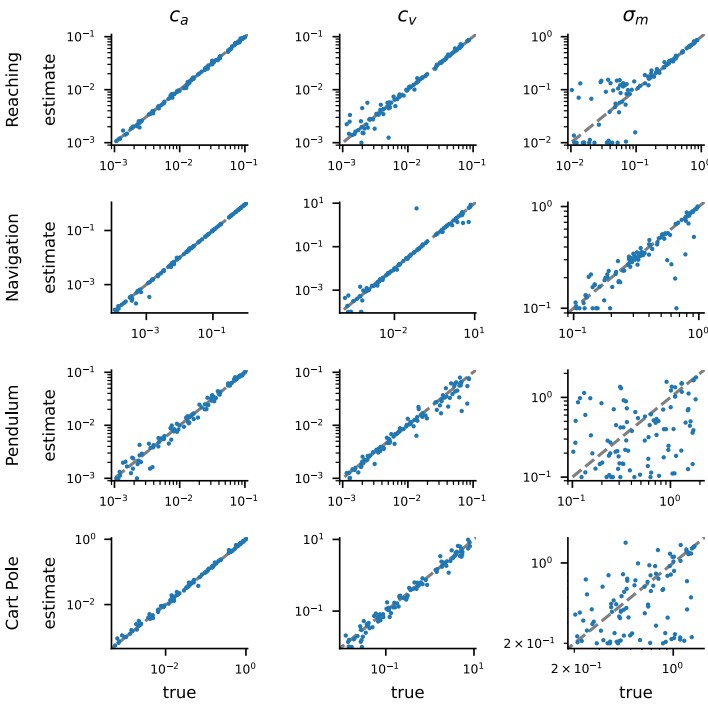

Figure S11: Maximum likelihood parameter estimates for the baseline with given control signals (fully observable)

## K.4 Quantitative results for the light-dark domain

As for the other tasks, we generated 100 sets of parameters sampled from uniform distributions in the following ranges

- horizontal target position $p$: $(-1, 1)$
- state-dependent cost parameter $c$: $(10^{-2}, 1)$
- perceptual uncertainty $\sigma$: $(10^{-2}, 1)$

and generated a dataset of 50 trajectories using iLQG [43] for each set of parameters. For each set, we then performed inference using our method and the baseline.

While the baseline accurately infers the target position, our method shows more variability in its estimates of the target position. The estimates of the cost term $c$ are comparable for both methods. However, the baseline completely fails to estimate the perceptual uncertainty, which our method manages to infer relatively accurately.

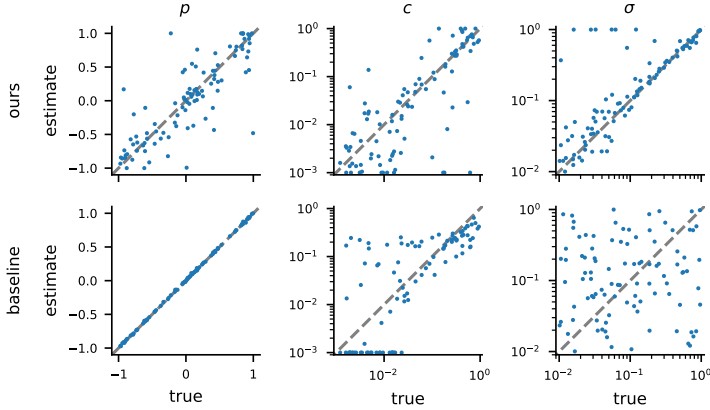

Figure S12: Maximum likelihood estimates for our method (top) and the baseline (bottom) in the light-dark domain.

