# OpenReview forum: "Probabilistic inverse optimal control for non-linear partially observable systems disentangles perceptual uncertainty and behavioral costs"
_NeurIPS.cc/2023/Conference — NeurIPS 2023 poster_

### Official Review · Reviewer_V7nV · 2023-06-27

**Soundness:** 3 good
**Presentation:** 2 fair
**Contribution:** 2 fair
**Rating:** 5
**Confidence:** 4

**Summary:**

This paper proposes an algorithm for inverse optimal control for agents operating in a partially observable Markov decision process, using only state trajectories. Given state trajectory data, dynamics model and observation model, the algorithm estimates the parameters by three steps: first, a policy is estimated using iLQG; second, the belief dynamics is estimated with EKF; finally, filtering is done by linearizing belief propagation. The likelihood is then maximized by back-propagation. The authors tested the proposed algorithm on several synthetic problems and compare it with a baseline based on maximum causal entropy. The results show that the proposed approach can better estimate the unknown parameters from behaviors than the baseline.

**Strengths:**

The problem studied here is relevant to community. I think especially that the authors consider the case without action information, which makes the setup more practical. The writing of the paper is easy to follow. The introduction is well motivated and the related work discussion is informative. The proposed idea is clean and is new to my knowledge. The authors discuss in details about the limitation of this work.

**Weaknesses:**

While the writing is easy to follow, it is unclear what the assumption is being made to use the proposed method. From Algorithm 1, the authors assume knowledge of the dynamics, observation models, and state trajectory, but the steps of Algorithm 1 requires more information than that, e.g., the cost is needed in iLQG. In addition, it is also unclear how the proposed algorithm actually operates; the authors describes how filtering and approximate likelihood (given all the parameters) is computed in details, but do not mention how the unknown parameters are actually estimated (the authors mention "using gradient-based optimization" with automatic differentiation, but it is unclear what the computational graph entails). As a result, while I get the high level idea, I do not think I fully understand the working details of the proposed scheme, and therefore it is hard to give it an accurate evaluation. I think the authors should compare more with recent IOC or IRL works, in the experiments, and consider more realistic datasets (rather than synthetic ones).  In addition, it would be more informative if the authors can better highlight their contribution in terms of the specific problem difficulties here (e.g., due to missing action).  Lastly, the proposed method is limited to low dim problems as the authors also point out.

**Questions:**

1. What is the assumption of known information made in the paper? It is not clearly specified in the current writing. From Algorithm 1, it seems that assuming only the dynamics and observation models, as well as state data trajectories. But in iLQG the cost is also needed, and in the EKF step the belief dynamics (i.e., beta_t) is needed. What are they in the experiments? The algorithm also uses the prior of belief. Is that assumed to be given too?

2. How does the learning actually work? The paper mentions the usage of automatic differentiation. But how is that actually done and what is the computation graph involved?  Algorithm 1 requires multiple linearization steps in iLQG, EKF, and the filtering. In particular, iLQG is an iterative process, do you also backpropagate through them?

3. While the paper supposes only state trajectories are given, it seems to make assumption on the knowledge of the belief dynamics, or what the belief of the agent is. It is not clear whether this is a stronger or a weaker assumption.

4. The current experiments only consider rather small and synthetic problems. Can you include results on some more realistic datasets? In addition, what are the agents that generate the data in the experiment (I can only tell iLQG is used for the light-dark domain)? It would be also good to test the proposed algorithm the estimation of non-iLQG agents, since the proposed one is based on iLQG, so that we can know the generality of the proposal.

**Limitations:**

Yes, the authors made clear what most limitations are. It would be also good to point out the limitation due to limited empirical evaluations done in the paper (e.g., due to the synthetic, low-dim nature, the agent policy's form, the parametric structure, etc.)

---

> ### Author Rebuttal · Authors · 2023-08-09
>
> Thank you for the generally positive assessment of our work. Below, we expand on some of the assumptions of our method. The questions raised about these assumptions will help us improve the clarity of our paper.
>
> Regarding the weakness point that a comparison with “more with recent IOC or IRL works” might be helpful, we want to highlight that the baseline we consider represents current state-of-the-art methods. Newer variants use mainly more expressive (deep) function approximations, which can learn more complicated cost functions in higher-dimensional spaces and with potentially unknown dynamics. For the setting we consider in our work, these methods, however, do not yield any advantage over the considered baseline let alone that they do not provide easily interpretable parameters that can easily be connected back to psychologically meaningful quantities.
>
> Questions:
>
> [Q1+Q2 What are the assumptions and how does learning work?] For our method, we assume that we are given a parametric form of the system dynamics, belief dynamics, cost function and initial belief of the agent which all might depend on unknown parameters that we would like to infer. Further we assume a set of state trajectories. You are right that for evaluating the likelihood (and in iLQG), one needs the values of the parameters, e.g., for the cost function. In our IOC method, the goal is to maximize the likelihood w.r.t. these parameters. To do so, we use a gradient-based optimization approach, i.e., we start with some initial parameters and differentiate using automatic differentiation through the likelihood computation (Algorithm 1).
> It is true that the iterative nature of iLQG significantly complicates the computation graph, because, naively, one would have to backpropagate through the iterative procedure, as correctly stated by the reviewer. We get around this by only linearizing once around the actually observed state trajectory (see Section 3.3), making the computation of the likelihood function and its gradients much more efficient.
> We then use the computed gradient to adjust our previous estimate in the direction of the optimal one.
>
> [Q3 Knowledge of agent’s belief] We would like to highlight again, that we do not assume knowledge of the agent’s belief. As shown in Fig. 1, the agent’s belief is an unobserved variable from the researcher’s perspective. What we assume is that there is a parametric model that describes how the agent’s beliefs are generated from the previous belief and observation (i.e. the belief dynamics). The beliefs themselves remain latent and need to be marginalized over using our probabilistic belief tracking formulation. Therefore, this is a weaker assumption than requiring the beliefs (which are internal to the agent in general) to be known.
>
> [Q4 Small and synthetic experiments] We agree with the reviewer that our method does not focus on high-dimensional problems, but we would like to emphasize that the tasks we consider are highly relevant in cognitive science and motor control, where relatively low-dimensional models are adequate for moving from controlled experiments towards naturalistic behavior. For example, the non-linear reaching model we consider (or similar models) has been widely applied in the neuroscience literature for modeling reaching data [Liu & Todorov, 2007; Knill et al., 2011; Wochner et al., 2020] and constitutes therefore a “realistic” task. Similar optimal control models have been employed in human locomotion [Papadopoulos et al., 2016] navigation [Kessler et al., 2022] or ball catching [Belousov et al., 2016].
>
> [Q4 Non-iLQG agents] The probabilistic formulation of the IOC problem (Section 3.1) is sufficiently general that it should be applicable to a broad class of agents beyond iLQG. In this paper, we evaluate it for iLQG agents, as (1) it allows efficient computation via local linearization (Section 3.2) and (2) many problems relevant in cognitive science and sensorimotor neuroscience can be solved via iLQG. For efficient and successful application with non-iLQG agents we would assume that further adaptation is necessary work and we therefore leave it as future work. However, we are convinced that laying out this conceptual framework is already a substantial contribution to the cognitive science and sensorimotor neuroscience community.
>
> We will make sure to include the mentioned limitation into our discussion in the revised version of the manuscript.

---

> > ### Comment · Reviewer_V7nV · 2023-08-20
> >
> > Thank you for your response. They address my concerns.

---

### Official Review · Reviewer_McGM · 2023-07-05

**Soundness:** 3 good
**Presentation:** 3 good
**Contribution:** 3 good
**Rating:** 6
**Confidence:** 1

**Summary:**

The paper introduces a new approach to inverse optimal control which is able to deal with partially observable systems in which action signals are not known. Most existing approaches only work in fully observable systems where actions are known. The paper introduces a probabilistic formulation for inverse optimal control and uses maximum likelihood to estimate the costs and parameters of the system. To make the likelihood tractable, they use local linearization similar to iLQG. The paper tests the approach on 4 tasks and shows improved performance over a maximum causal entropy-based baseline.


**Strengths:**

I am not familiar with this area of research and most of the concepts in the paper are new to me but I was still able to follow most of concepts introduced in the paper hence I think the writing is mostly coherent and fairly easy to follow.


**Weaknesses:**

I don't have any weaknesses to point, I mainly have questions which I will write below.


**Questions:**

1) My first question is about baselines. The authors use a maximum causal entropy-based baseline for the experiments. From my understanding, maximum causal entropy is mainly used for exploration in RL but it is not clear to me what is the IOC-based method used in MCE to estimate the cost function. Does it use something like ILQG or EKF?
2) What functions are used to model f and \beta in equation 2? From my understanding, the functional form should be known beforehand which would be quite limiting for a setting where this information is not known.


**Limitations:**

The authors have discussed all relevant limitations.

---

> ### Author Rebuttal · Authors · 2023-08-09
>
> Thank you for rating our paper as easy to follow, even for a reader outside this specific area of research! The raised questions, which we answer below, will help us improve the accessibility of our paper even further.
> As the reviewer stated, that he has limited familiarity with the field, we would like to refer to the significance statement that we included at the beginning of this rebuttal.
>
> Questions:
>
> [Q1: Baselines] Yes, that is correct, MCE controllers add variability to the actions and are therefore often used for exploration. For the inverse problem, one usually assumes that the expert is not perfectly accurate and therefore an MCE controller is often used to model the variability of the agent. The MCE formulation for the inverse problem is also of a suitable mathematical form, making it efficiently applicable. As the IOC problem based on MCE cannot be solved exactly, there have been various approximations introduced, partly based on linearization [e.g., Levine and Koltun, 2012], making the application with iLQG straightforward. Specifically, in contrast to our method, there is no explicit model of partial observability, and action signals are assumed to be known. There has been very little work on IOC for partially-observable systems (see related work) and to the best of our knowledge, the EKF has not been used with the MCE formulation in IOC. Thanks for raising this point about the baseline, which we will clarify in the revised version.
>
> [Q2: Functional form of (belief) dynamics] It is true that the functional form of the system / belief dynamics is assumed to be known beforehand. This is typical in the application domains we have in mind, i.e. cognitive science and neuroscience, where researchers have quite accurate models of the sensorimotor system and are interested in inferring parameters of this system (e.g. costs, uncertainties, etc.). While more general function approximators, e.g. neural networks, which are also parametric models (albeit with more parameters) could be used for the dynamics, we are interested in examples with an interpretable (and therefore often low-dimensional) parameter space.

---

### Official Review · Reviewer_xp9a · 2023-07-07

**Soundness:** 3 good
**Presentation:** 3 good
**Contribution:** 3 good
**Rating:** 7
**Confidence:** 4

**Summary:**

The authors propose an inverse optimal control method to handle the challenging case of inferring an internal model in a non-linear partially observable system, when the action sequence is not observed. Quantitative evaluation of the proposed method was shown for some classic control problems. The authors also demonstrated through an example the potential of their method in disentangling perceptual factors and behavioral costs.

**Strengths:**

The written language is easy to understand, and well organized. This work is a novel combination of ideas from some well-established techniques. The method should be of particular interest since intended actions of other agents are often only partially observable.

The value of the method is highlighted by a comparison with an alternative method that mistakenly attributes perceptual uncertainty to subjective preferences. The authors’ IOC is able to correctly infer that much of the action in this task is driven by exploration rather than exploitation.


**Weaknesses:**

It would be helpful to highlight more strongly that they can distinguish preferences from uncertainty. The authors rightly mention this virtue in the title, so I think it deserves more attention in the text. On the other hand, for that section, I would say that stronger evidence is needed, for example on a more complex task than the light and dark one. Uncertainty and cost are sometimes nonidentifiable: It could be too costly to correct a certain mistake, or it could be too uncertain to justify an optimal action, and these effects can be indistinguishable from the outside. However, when the uncertainty is dynamic while the cost function is static, these factors are dissociable. This should be addressed by the authors.

**Questions:**

The authors describe their method as uniquely accounting for unobserved actions. But other Inverse Optimal Control methods allow for stochastic policies [e.g. Wu, Kwon et al 2020], in which actions can equivalently be interpreted as a latent intended action distorted by noise, or as a draw from a known policy. Aren’t these equivalent if the policy is optimized with action noise?

The authors should show a sanity check that their linearization successfully allows us to estimate actions near enough the ground truth.

It would be helpful to discuss more of the limitations from combining iLQG with EKF. Do the authors allow for controls that depend on a dynamic covariance, or just estimates of the mean? This was a bit unclear from their writing. It’s likely, given the techniques, that their method does not account for controllable uncertainty. Is this correct?

I would like to see more evidence for the main claim, as suggested in the title, that the method disentangles perceptual uncertainty and behavioral costs. To show that taking into account the agent’s beliefs helps, it would be helpful to compute how uncertain the internal belief is while the agent moves towards the light before turning towards target, and plotting how it changes with subjective reward parameter c.

**Limitations:**

The authors have a thoughtful discussion of limitations.

---

> ### Author Rebuttal · Authors · 2023-08-09
>
> Thank you for the positive review, and specifically for highlighting the importance of disentangling costs from uncertainty. We will use the additional page of the camera-ready version to expand on this in the introduction to make it a more prominent feature of our work. The issue of potential non-identifiability of uncertainty and costs in certain cases is also a good point, that of course applies to each and every algorithm in IRL and IOC. We agree that in general, there can be cases like the one described in the review, where these factors cannot be distinguished. We will address this in the discussion. Our goal was to show that across several example scenarios, including those common in the motor control and cognitive science literature, where the costs and uncertainties can be disentangled, our method is able to do this while the baseline is not. Thus, our approach is a step towards starting to explore these issues of model identifiability.
>
> Questions:
>
> Unobserved actions (in comparison to Wu, Kwon et al., 2020): In their PNAS paper, Wu et al. (2020) assume both the agent’s observations and the agent’s action signals to be fully known (see their figure 1B, where both the observation and the action are indicated as observed variables from the researcher’s point of view). In their NeurIPS paper, Kwon et al. (2020) assume the action signals to be known, but marginalize over the agent’s internal observations / belief (see their Figure 1 and algorithm 2). So, both of their formulations assume observed action signals, in contrast to our method, and additionally assume a stationary policy. But yes, they also use a stochastic policy in addition to the partial observability formulation - even if it does not correspond to the MCE controller widely used.
>
> Sanity check of linearization: Thank you for your suggestion to show “a sanity check that their linearization successfully allows us to estimate actions near enough the ground truth”. Could you elaborate a bit further on what you precisely mean by this? The parameters that are estimated are close to ground truth, see figures 2 & 3 and appendix I, J, and K. The question asked about actions, though. We estimate actions only for determining the linearization points (and for the baseline) by using the non-linear system dynamics (appendix G). We do not claim that these estimates are very accurate as they ignore noise (and the baseline would perform better then) but are sufficiently close to yielding appropriate linearization points.
>
> Limitations of combining iLQG with EKF: It is true that the combination of iLQG and EKF results in controls based only on the mean of the belief. However, the partially-observable version of iLQG (Li & Todorov, 2007), which we use here, shows uncertainty-aware behavior (as we show in the light-dark domain), because the state-dependent covariance is taken into account during the computation of the control law. In general, an extension to control methods that are based on the covariance of the belief (e.g. belief-space iLQG, van den Berg et al., 2012) should be possible in our framework, by defining the belief dynamics based on a vector including the covariance. This extension is a fruitful idea for future work and will be added to the discussion.
>
> [More evidence that the method disentangles uncertainty and costs] Thanks for raising this point. We should have been clearer about the experiment in the light-dark domain. To illustrate sources of information-seeking behavior, we have created an additional figure that shows the agent’s belief for varying cost parameters (Figure 1 in the additional pdf). An agent who takes uncertainty of the belief into account during computation of the control law (e.g. partially observable iLQG; Li & Todorov, 2007) will move towards the light source before approaching the target. Adding another cost term (c) that expresses a preference to be close to the light source does not change much about this behavior (top row). An agent who computes the control law irrespective of the belief uncertainty (e.g. fully observable iLQG; Todorov & Li, 2005) will not move towards the light source per default. As a result, the belief uncertainty is higher. Only when we add an extra term in the cost function (c) to force the agent to go to the light source does the agent move to the right before approaching the target. These two sources of information-seeking behavior can lead to similar trajectories. To show that we can disentangle these factors, we simulated 100 random parameter sets (varying the target position $p$, perceptual uncertainty $\sigma$ and cost $c$). In Appendix K.4, we show that our method can infer all of these parameters, while the baseline cannot infer the perceptual uncertainty. We will add the aggregated results of the light-dark problem to Fig. 3 of the final version of the paper.

---

### Official Review · Reviewer_CoMo · 2023-07-21

**Soundness:** 3 good
**Presentation:** 3 good
**Contribution:** 2 fair
**Rating:** 5
**Confidence:** 3

**Summary:**

This paper targets for solving the inverse optimal control problem for partially-observable stochastic non-linear dynamics with no observation of the action. To estimate the parameters of the cost function in a stochastic non-linear system, the author first derives a likelihood function for the model parameters. They then approximate this likelihood by locally linearizing the system using a combination of iterative linear quadratic Gaussian (iLQG) and Extended Kalman filter (EKF) techniques. The proposed inverse method makes explicit assumptions about the dynamics of the control tasks, the agent's belief in a stochastic environment, and the structure of the cost function. The authors evaluate their approach on four simulation environments where structured noise is added to the dynamics of the environment to introduce stochasticity into the system dynamics. In comparison to the baseline, the proposed approach demonstrates better uncertainty estimation.

**Strengths:**

In general, I find this paper easy to follow and variables in the equations are well defined. The derivations are nicely structured. The details are well documented in the Appendix and the assumptions and limitations are explicitly considered.


**Weaknesses:**

I suggest the author clearly define their contribution in this paper, as the likelihood formulation of the inverse optimal control problem has been derived in another work, \textit{Inverse Optimal Control Adapted to the Noise Characteristics of the Human Sensorimotor System}. To enhance clarity, I recommend placing the likelihood formulation in the background section.

Claiming that their approach can disentangle perceptual uncertainty and behavior costs might be too bold for two reasons: firstly, the author demonstrates this claim in a single simulation environment only, and secondly, the structure of perception uncertainty and behavior cost function is known. However, in reality, obtaining the structure of these elements is difficult.

My major concern pertains to the number of assumptions made in different aspects of the system to obtain the proposed closed-form solution. For instance, all noises added in the experiments are assumed to be Gaussian distributed. These assumptions render the proposed approach challenging to apply in real-world scenarios, and the experiments were solely conducted in simulations. In addition, potential solutions to these assumptions i.e. addressing multi-modals belief utilizing SMC, are only briefly mentioned in the conclusion section.


**Questions:**

In the reaching task and the navigation task, why is the goal position not inferred?

In the conclusion, the authors mentioned that their method may not be effective for high-dimensional parameter spaces. It would be beneficial for the community to know the accuracy of the uncertainty estimation as the number and dimension of parameters scale.

In the sensorimotor domain, do you always have a well-structured cost function that you can obtain beforehand? I.e. I think your approach would not work if the formulation of the behavior cost function like eq.8 is not given.


**Limitations:**

Limitations are well documented in the paper.

---

> ### Author Rebuttal · Authors · 2023-08-09
>
> Thank you for the generally favorable evaluation of our paper.
>
> Weaknesses:
>
> Contribution: Yes, the likelihood formulation has previously been derived for linear systems in that paper. Here, we write it in a more general form, which is applicable beyond linear systems, which is why we kept it in Section 3. Importantly, we derive new algorithms to actually carry out the inference for the case of nonlinear dynamics, nonlinear observations, including much more general noise distributions. This was previously unavailable.
>
> Disentangling uncertainty and costs: One cannot build scientific models without making assumptions. For example, without any knowledge about parametrizations of perceptual uncertainty and parametrizations of the cost function, it is not possible to unambiguously identify these. We completely agree that obtaining the structure of perceptual uncertainty and behavioral cost from behavior is a very difficult problem. In fact, our current work is to our knowledge the only one for the case of nonlinear dynamics, nonlinear observations, interesting noises and unobserved actions. Therefore, we argue that explicitly stating the assumptions, under which conclusions about these factors are drawn, is a good thing. Here, we provide a framework for spelling out such assumptions in the form of parametric models and inferring their parameters. In many tasks in cognitive science and neuroscience, one has knowledge about some part of the model, e.g., the system dynamics or belief dynamics up to certain parameters but other quantities, e.g., costs and perceptual uncertainties are latent.
>
> Assumptions about the system: It is true that the noises v_t and w_t that go into the dynamics and observation function are Gaussian. However, they are transformed within these functions, resulting in non-linear signal-dependent noise models. This assumption is motivated by what is known about the sensorimotor system (e.g. Harris & Wolpert, 1995; Todorov and Jordan, 2002). Thus, very complex noise distributions can be handled by the current model.
>
> Questions:
>
> Why is the goal position not inferred? In reaching and navigation tasks, the goal position is usually known to both the agent and the experimenter, which is why there is no need to infer it. The (internal) cost and perceptual uncertainty, on the other hand, can usually not be set or determined by the experimenter. Further, we aimed to keep the parameters consistent across tasks, and therefore used a similar parameterization for all of them. In principle, the goal position could be inferred easily using our method as long as the problem remains uniquely identifiable.
>
> Scaling with number of parameters: Thank you for your suggestion, this would be indeed interesting. However, we do not believe that there is a general scaling law, as the estimation depends on the specifics of the problem more than on the mere number of parameters.
>
> Structure of the cost function: In a motor control experiment, researchers typically have a good idea of the task being performed by the agent and potentially other factors influencing performance (cognitive, biomechanical effort costs, etc.), which can be formalized as parameters of a cost function. For example, the costs of movements may depend on path length, accelerations, or torques of movements.  If such a parametric cost function is not available, one would have to resort to more general function approximation methods, which can be more challenging to render interpretable.

---

> > ### Comment · Reviewer_CoMo · 2023-08-16
> >
> > Thank you for the comments. After reviewing your response to reviewer V7nV, the assumptions made in your approach are much clearer to me. I suggest that you also include the statement about these assumptions in your revision. It looks to me your approach could be beneficial to your particular domain while given somewhat limited contribution as mentioned by other reviewers, so I will update my score to 5.

---

### Official Review · Reviewer_1CfE · 2023-07-22

**Soundness:** 3 good
**Presentation:** 4 excellent
**Contribution:** 3 good
**Rating:** 6
**Confidence:** 3

**Summary:**

This paper introduces a probabilistic approach to inverse optimal control for partially-observable stochastic non-linear systems with unobserved action signals. It derives an approximate likelihood function for the model parameters by linearizing the system around the observed trajectories and tracking the agent's belief distribution. This method is demonstrated to accurately infer the parameters better than the baseline maximum causal entropy (MCE) approach on two classic control tasks and two human behavioral tasks. Additionally, it shows the ability to disentangle the influences of perceptual uncertainty and behavioral costs on information-seeking behavior sources of information-seeking behavior.

**Strengths:**

- The introduction gives a clear background, which is helpful for someone like me who is less familiar with this field. It provides a clear understanding of the research question and the existing approaches.
- This paper proposes a new probabilistic approach for inverse optimal control in stochastic non-linear systems with missing control signals and partial observability, which outperforms the baseline MCE model and also provides an interpretable representation of the parameter space.
- The proposed method is evaluated on different tasks, including two human behavioral tasks, which might inspire further investigation on how the method can be applied to neuroscience and cognitive science studies.

**Weaknesses:**

A more comprehensive investigation into the method's robustness under various conditions would strengthen the paper's contributions.

**Questions:**

Why is it necessary to assess the proposed method's performance on pendulum and cart pole tasks, despite their deterministic nature and lack of partial observability?

**Limitations:**

The authors have adequately acknowledged and addressed the limitations of their methods, which include the following: (1) The focus on tasks that can be well-solved by control methods based on linearization and Gaussian approximation (iLQG and EKF), which may not fully capture the complexity of naturalistic behavior, and (2) the concern that the method might not scale effectively to high-dimensional parameter spaces as optimization in a high-dimensional non-linear space can potentially lead to getting stuck in local minima.

---

> ### Author Rebuttal · Authors · 2023-08-09
>
> We would like to thank the reviewer for the positive assessment of our work.
>
> More comprehensive investigation into the method’s robustness: We agree that, as always, there is also in this manuscript room for even more evaluations. Figure 3 gives evaluations involving 100 datasets each for the considered problems. Additionally, appendix I, J, and K provide more evaluations.
>
> Evaluation on pendulum and cart pole: While the pendulum and cart pole tasks are deterministic and fully observable in their standard versions, we have implemented stochastic and partially observable versions of these tasks. While our main focus of our method is the application in cognitive science and motor science, we used these tasks to show that our method is applicable to this domain but can find applications also in other fields, such as robotics, where partial observability and stochasticity might also play a role.
>
> We also agree that in principle much more naturalistic tasks are conceivable and indeed, over the last 15 years we have continuously contributed to establishing more naturalistic tasks in the field. However, over the last few years, probabilistic methods have found wide acclaim, including at NeurIPS, that do dynamic state inference for tasks that are more naturalistic but not fully unconstrained, such as navigation. Here, we provide not only inference of the belief state but additionally inference of the costs, and the uncertainties, which includes disambiguating pragmatic from epistemic actions. Thus, we think that this is a significant contribution to the field, it extends current analyses, and should be widely applicable in the field.
>
> If these are the only two concerns with our paper, we would like to ask the reviewer politely whether increasing the score is appropriate.

---

### Official Review · Reviewer_ff8k · 2023-07-24

**Soundness:** 4 excellent
**Presentation:** 2 fair
**Contribution:** 3 good
**Rating:** 7
**Confidence:** 2

**Summary:**

This paper is a strong contribution on the Inverse Optimal Control problem when actions cannot be observed. It is particularly interesting their modelling of the agent and the “researcher” observer. The mathematical depth is good and sound. The results may be enough for a theoretical paper. However, there are too many unknowns to be ready for acceptance, particularly some definitions like partial observability related to noise. Another issue is the clarity regarding the baseline used, is it self-programmed?, is it taken from previous literature?. It seems that it does not work for any of the problems tested.

**Strengths:**

-	The mathematical depth is good.
-	Inverse optimal control plus noise estimation is relevant for many applications; especially for motor control research.
-	Results may be enough for a theoretical paper.


**Weaknesses:**

-	Narrative: While the contribution looks very promising, the explanation of the contribution in the introduction is not enough clear to understand what is the main focus of the paper. Furthermore, there is a need to jump forward and backward in the text to fully understand the details of the approach. Particularly, it is sometimes complicated to see if the authors are talking about the agent or the observer (“researcher”). Another example: I had to read the whole article to understand the title.
-	The partial-observability may be controversial as it is defined. The idea of tracking the belief of the agent is interesting, but why not using directly the observed state? So then is it the noise that the algorithm is tackling as partial observability. Thus, then the complexity depends on the type of noise.
-	Following the previous comment, while the method may be used for any parameter estimation (as authors state) why only estimating the observation/motor noise? Btw, observation noise of the agent or the observer? From the text I assumed that the authors were doing system identification at the same time. But they are estimating the noise.
-	Baseline explanation needs further description for the sake of clarity.


**Questions:**

-	The description of the variables v_t and w_t could be improved
-	Why there is a final cost that is independent on the action taken? Conversely to the previous time steps.
-	What is πt(xt) ? (section 2.2)
-	Why stochastic policy has a capital Pi?
-	Not clear why the EKF is introduced.
-	“In the inverse optimal control problem, the goal is to estimate parameters θ ∈ Rp of the agent’s optimal control problem given the model and trajectory data. These parameters can include properties of the agent’s cost function, the sensory and control systems of the agent, or the system’s dynamics.” This is a definition proposed by the authors. Originally, IOC is to recover the cost function from expert demonstrations. Getting system parameters is called system identification.
-	“cost of final velocity cv” Is this related to the cost definition with the final cost at end time?
-	“observation noise” of what quantity? Experimenter, agent?
-	“Note that past methods based on MCE are usually limited to estimating cost functions, so that parameters such as the agent’s noise therefore cannot be inferred.” This sentence helps a lot to understand the authors approach.
-	It is not clear why the dynamics of the agent state and its beliefs only depends on the state and the belief: “closed-form expression for p(xt+1, bt+1 | xt, bt).”
-	“we applied a baseline method based on the maximum causal entropy (MCE) approach [3].” Which method? Self-programmed or taken from the literature? Baseline text could be improved. In the end I am not sure how many baselines are being used.
-	“expressed as a non-quadratic cost function of the joint angles” Is this really true. Besides, while I understand that there is not enough space for everything on the paper, the description of the input output system would be nice to place it in the main paper so there is no need to go to the appendix to read that the arm is controlled with torque.
-	It is not clear why the baseline cannot even recover the action cost properly. Would it work in the absence of noise?
- The agent navigation dynamics can be simplified to use the turn rate as the control action:
[dot{x}, dot{y}, dot{theta}]¨=  [cos(theta)w, sin(theta)w, u]



**Limitations:**

Limitations are sufficiently addressed. My only constructive suggestion is to mention that noise covariance estimation for non white-noise sources is a very important problem in control. So I would include it in the discussion.

---

> ### Author Rebuttal · Authors · 2023-08-09
>
> We thank the reviewer for acknowledging the strong theoretical contribution of our work. The many detailed questions, which we answer below, will help us improve the clarity of our paper.
>
> [Noise variables] The noise variables $v_t$ and $w_t$ are both standard Gaussian random vectors. However, this does not mean that only additive Gaussian noise is possible. These variables can be subject to potentially non-linear transformations in the dynamics and observation functions, resulting, e.g., in signal-dependent noise models. Therefore, this accommodates both Weber-Fechner-type sensory uncertainty in humans and animals as well as signal-dependent noise of motor actions in humans and animals.
>
> [Final cost] This is a standard formulation in finite-horizon control problems (see e.g. Wikipedia on LQG). For time 1 up to T-1, the agent takes an action and receives a reward based on the current state and the action. At time T, the agent reaches the final state and can no longer perform actions. Therefore, the final cost depends only on the final state.
>
> [$\pi_t(x_t)$] This is the agent’s policy function, which maps the current (estimate of the) state to the action. Note that later in the paper, we use a probabilistic policy, so that $\pi_t$ also depends on a noise variable.
>
> [Capital pi] For the stochastic policy, we denote with $\pi_t$ the policy function, depending besides the belief also on the noises. We use $\Pi_t$ to denote the resulting policy distribution, just depending on the belief.
>
> [EKF] The agent has sensory uncertainty about the true state of the world (partially observable MDP), different from the RL setting. We use the EKF for modeling the belief dynamics of the agent, so it is used in our algorithm. Please refer to Section 3.2 to see how the concrete choices for belief dynamics (EKF) and policy (iLQG) are used in our IOC algorithm.
>
> [IOC vs. SI] You are right that commonly IOC is used for inferring parameters of the cost function given behavioral trajectory data. System identification, on the other hand, usually denotes the whole process of learning the dynamics of the system and includes determining controls for the data generation as well. Note, however, that in motor control, neuroscience, and cognitive science, the system is usually well characterized, e.g. muscle dynamics or the kinematic chain of a lib. As we consider the task of inferring the parameters of the system given a fixed set of trajectories, from our point of view, this corresponds best to the setting of IOC, which is why we labeled it this way.
>
> [Final velocity]: In our considered tasks, the system should be controlled to have close to zero velocity at the final time step. The parameter $c_v$ determines how high the cost for the velocity penalty at the final time step is. More details and precise formulas of the cost function parameterizations are provided in appendix J.
>
> [Obs. noise] We assume that the researcher can observe the state and the agent has noisy/partial state observations (Fig. 1 / section 3). Thus, the observation noise is regarding the agent.
>
> [State and belief dynamics] Generally, the state $x_{t+1} = f(x_t, u_t, v_t)$ in a POMDP can depend on all previous observations of the agent $y_{1:t-1}$, because the agent can choose an action based on all prior observations. One can simplify the problem by assuming that the agent’s action $u_t = \pi(b_t, \xi_t)$ is a function of the agent’s current belief (and perhaps some noise). The belief, in turn, is a function of the previous belief and observation $b_t = \beta(b_{t-1}, y_{t-1})$. This allows us to write a system of states and beliefs, which only depends on the previous state and belief. We adapted this idea from van den Berg et al. (2011).
>
> [Which baseline] As previously stated, more details of the baseline can be found in appendix B. We are not aware of any previous approach that shares the same setting, i.e. partial observability of states and actions, and therefore, past implementations cannot be directly applied to the setting we consider (e.g. finite time horizon). Accordingly, we implemented the baseline ourselves. As there exist various approaches on how to choose a feasible approximation, we show that our implemented baseline works for the regarded tasks in the usual setting of IOC (fully-observability and given control signals) in appendix K3.
>
> [Cost] Good point, we should have mentioned that the reaching task is torque-controlled, which considerably complicates both the control and the IOC. Because the representation of the state consists of elbow and shoulder angles and the cost function depends quadratically on the hand position, which is a non-linear function of the angles, the cost is a non-quadratic function of the state. We will use the additional page of the camera-ready version to clarify this by expanding the description of the reaching task in the main text and adding some of the equations from the Appendix.
>
> [Baseline cost] The main reason why the baseline cannot recover the cost is that it assumes the agent’s action signals are known, which is essentially the assumption of all previous methods that we are aware of. When we provide the baseline with the action signals, it can recover cost parameters, but still struggles with noise parameters (see Appendix K.3). In the absence of noise, the baseline would work well, as the true actions could be exactly estimated based on the states.
>
> [Agent navigation dynamics] We politely disagree with the reviewer that the agent navigation dynamics can be simplified to the proposed form. The proposed model is actually very similar to the dynamics model we are using, but we additionally allow the agent to change the forward velocity $w$ by adding an acceleration to it, which is why we include it in the state. Please also keep in mind that we are using discrete time (difference) equations instead of continuous time (differential) equations, which also account for a difference in notation.

---

> > ### Author Response · Authors · 2023-08-10
> > **Additional comments on weaknesses**
> >
> > [Explanation of contribution] To summarize our contribution in one sentence, we introduce an inverse optimal control method that can deal with partially observable stochastic systems, where the agent’s action signals are unknown to the researcher. This requires distinguishing between the control problem from the agent’s viewpoint (Fig. 1A) and the inference problem from the researcher’s viewpoint (Fig. 1B). We have tried to be precise in our wording and always clearly distinguish these two perspectives, but we are happy to elaborate if any concrete passages in the paper are unclear. As for the title, we think that the title both captures the problem setting (stochastic partially observable systems) and the unique selling point (disentangling perceptual uncertainty and behavioral costs).  See also the answer to all reviewers at the beginning of this rebuttal.
> >
> > [Partial observability] In IOC we take the perspective of the researcher, who observes a trajectory of states $x_t$, but does not have access to the agent’s noisy observations $y_t$ (see Fig. 1 in the paper). The agent cannot directly observe the true states $x_t$, but instead receives noisy / partial observations $y_t$. Based on these observations, the agent forms a belief state $b_t$ (e.g. using the EKF). We as researchers cannot observe this belief state, because it is a quantity internal to the agent. The likelihood $p(x_1:T)$ depends on the true states $x_t$, which are observed by the researcher. To compute this likelihood, the unobserved internal variables of the agent need to be marginalized out. Although this way of formalizing the IOC problem might be novel, we do not think it is controversial.
> >
> > [Observation noise] We infer parameters of the cost function and of the noise model. For inferring the parameters correctly, the problem still needs to be uniquely identifiable. For the regarded problems, it is therefore not possible to do full system identification while estimating the cost function, as the problem quickly becomes highly unidentifiable. We therefore limit our evaluation to problems with few interpretable parameters that are expected to be uniquely identifiable. Experiments in cognitive and motor science are usually designed this way. Importantly, dynamics in these settings are usually well characterized, e.g. derived from first principles such as kinematics or measured empirically in separate experiments.
> >
> > [Baseline description] For the sake of readability, we preferred to have a rather short intuitive explanation of the baseline in the main text. In Appendix B, there is a detailed and more formal description, which should contain all necessary descriptions. If there are any further concrete questions about the baseline, we are happy to answer them and improve our description.

---

> > ### Comment · Reviewer_ff8k · 2023-08-15
> > **Thank you for the clarification**
> >
> > Thanks for the comments. I really think that the paper can be improve the clarity of the presentation. Particularly, being clear about partial observability vs noise so the experiments contribution is clear. I would reread the paper in detail with the new comments and make my final evaluation.

---

> > > ### Comment · Reviewer_ff8k · 2023-08-16
> > > **Final comment and additional suggestion**
> > >
> > > I think this is an interesting contribution, thus I would rise the score. You can consider in the future replacing the EKF with new forms of probabilistic filtering, such as Dynamic Expectation Maximization. You can even learn the covariance matrixes online: See Meera 2023 https://arxiv.org/abs/2308.07797.

---

### Official Review · Reviewer_EbpC · 2023-07-26

**Soundness:** 3 good
**Presentation:** 3 good
**Contribution:** 2 fair
**Rating:** 4
**Confidence:** 2

**Summary:**

This paper presents the new formalization of inverse optimal control on partially-overevable Markov decision processes. The authors argue that most existing works on inverse control or inverse reinforcement learning focus on fully-observable Markov decision processes. Their approach extends iterative linear quadratic Gaussian (iLQG) and Maximum causal entropy (MCE) reinforcement learning, introducing local mineralization to achieve tractable likelihood. The method is evaluated through simulations.


**Strengths:**

+ The formulation and derivation of the method are solid and well-motivated.

*Weaknesses




*Limitations


**Weaknesses:**

- The derived method seems natural, making it challenging to identify the novelty of the proposal.
- The evaluation lacks sufficient quantitative and qualitative analysis, as it only covers simple settings and compares with the MCE approach ~~without using the proposed method~~.


**Questions:**

1. Considering the numerous studies on inverse reinforcement learning and inverse optimal control without the assumption of linearization of the dynamical system and Gaussian approximations, a justification is needed in terms of applicability to robot learning. A comparison with methods without such assumptions should be included. Can the method handle more dynamic tasks such as walking?

2. Clarify the difficulty of the problem in a more intuitive way to explain why this problem remains unsolved. The formulation is somewhat straightforward. Therefore, potential readers may wonder why the problem is still unsolved.

**Limitations:**

This is just a suggestion.

Regarding POMDP in robot learning, exploring the relationship with world models (e.g., [1,2]) would be valuable.

[1] Hafner, Danijar, et al. "Dream to control: Learning behaviors by latent imagination." arXiv preprint arXiv:1912.01603 (2019).
[2] Tadahiro Taniguchi, Shingo Murata, Masahiro Suzuki, Dimitri Ognibene, Pablo Lanillos, Emre Ugur, Lorenzo Jamone, Tomoaki Nakamura, Alejandra Ciria, Bruno Lara & Giovanni Pezzulo (2023) "World models and predictive coding for cognitive and developmental robotics: frontiers and challenges," Advanced Robotics, 37:13, 780-806

---

> ### Author Rebuttal · Authors · 2023-08-09
>
> We thank the reviewer for calling our formulation and derivations solid and well-motivated. In our answers to the specific questions below, we elaborate on the perceived lack of novelty and simplicity of our evaluations.
>
> We are a bit unclear about what is meant by the purported weakness that "The evaluation [...] compares with the MCE approach without using the proposed method." In our evaluation, we do use both the MCE method and our proposed method. Could you elaborate?
>
> Answers to questions:
>
> 1. We acknowledge that there are methods for IRL and IOC in settings with more complicated dynamics models such as walking. However, these methods typically work in deterministic (!), fully-observable (!) settings and assume the agent’s action signals to be observed (!) (i.e. most problems implemented in OpenAI gym and comparable frameworks). Here, we are motivated by problems relevant to cognitive science and neuroscience, where biological systems are often modeled as having noisy sensory and motor systems, resulting in stochastic, partially-observable problems with unobserved action signals. This complicates the inverse optimal control problem significantly. We present a method that explicitly models these factors and therefore achieves good results on a range of different problems, albeit with simpler dynamics compared to methods applied on deterministic fully-observed problems, which is to be expected. If a relevant scenario in robotics is of interest, it could be kinesthetic teaching or imitation learning. Our method e.g. allows taking into account the specific human signal-dependent motor variability or internal biomechanical cost.
>
> 2. Thank you for raising this point. We agree that an IOC formulation that derives the generative model for a given forward problem and inverts it using probabilistic inference might seem straightforward. We argue that this is a virtue of our probabilistic problem formulation instead of a shortcoming. We have included a significance statement at the beginning of our answer to all reviewers. We will make sure to include a precise statement about what current methods in IRL and IOC cannot do that our method achieves.
>
> Thanks for the suggested references about POMDPs in robot learning, which we happily will include. We agree that extending the proposed approach towards learning of world models is a promising direction, and we will include this in the discussion. However, they concern very different problems. Even if a world model is given, when observing a behaving agent, it is not clear what the internal belief states are, and what the internal cost functions are, particularly, when not observing the full description of the internally generated actions.

---

> > ### Comment · Reviewer_EbpC · 2023-08-15
> >
> > Thank you very much for your response, including the general comment to all reviewers. Now, your motivation and contribution have become clearer to me.
> >
> > >We are a bit unclear about what is meant by the purported weakness that "The evaluation [...] compares with the MCE approach without using the proposed method." In our evaluation, we do use both the MCE method and our proposed method. Could you elaborate?
> >
> > I apologize for causing confusion. Please disregard the phrase "without using the proposed method." I believe the authors have adequately explained their reasons for not incorporating more baseline methods.

---

### Official Review · Reviewer_TQks · 2023-07-28

**Soundness:** 2 fair
**Presentation:** 3 good
**Contribution:** 2 fair
**Rating:** 5
**Confidence:** 3

**Summary:**

In this paper, the authors propose a method to infer an agent’s internal model in a Partially Observable Markov Decision Process (POMDP) when the agent’s actions are non observable. Using local linearization, the authors show how a closed form approximation of the likelihood function for state trajectories can be constructed to subsequently yield maximum likelihood estimates. They also show that when there are confounding factors that can lead to the same behavior, the proposed method is able to disambiguate the factors better compared to the baseline.

**After author's rebuttal:** I appreciate the authors offering clarifications. My main concern was around the significance of the contribution, which I based on the references discussed in the paper since I haven't been working on this specific area myself. However, looking at the other reviews and the author's comments, it appears the paper does more than just relaxing a simple assumption or two. I also mentioned that if the inference mechanism is aware of the generative model, one would expect better estimation in general. I make this comment from a Bayesian perspective so if the observations are from a Gaussian distribution and we model it as such, we expect to learn the sensible parameters with enough data as opposed to an unaware model that assumes, say, a Laplace distribution. I would need to think more as to why this might not hold here, as I am still inclined to believe it does. In any case, based on the discussion of the contributions, I am changing my rating from 4 to 5.

**Strengths:**

The paper does a good job at laying down the groundwork for the problem, going over previous work and pointing at the potential shortcomings of existing methods. The motivating example in Fig. 1 works well and the paper’s structure follows naturally. The experiments are well-designed and the results are clearly discussed establishing the technique’s superior performance over the baseline.

**Weaknesses:**

However, in my opinion, the contribution lacks significance for acceptance at the venue. The problem formulation is slightly more general than existing work but only marginally so. The linearization to get approximate likelihood is clean but not novel. The results are not surprising since a technique that is aware of the generative assumptions is expected to lead to better posterior estimates. The disambiguation behavior follows from better estimation.

**Questions:**

NA

---

> ### Author Rebuttal · Authors · 2023-08-09
>
> We thank the reviewer for their positive assessment of our exposition of the problem, the experiments, and the results. However, we would like to ask the reviewer to substantiate the claim that “the contribution lacks significance for acceptance at the venue”. Looking at the publications at Neurips from recent years, IRL methods have been published that are targeted at inferring animal behavior completely excluding perception (partial observability due to sensory uncertainty), assuming the actions to be fully observed (the observed action is assumed to be identical to the planned and intended movement), and, therefore, cannot even accommodate the notion of actions being epistemic or pragmatic. Our problem formulation for partially observable systems with unobserved action signals, which the reviewer calls “slightly more general than existing work but only marginally so”, is motivated by a wide range of experiments in neuroscience and cognitive science. To the best of our knowledge, there had previously been no inverse optimal control method for these cases.
>
> We also do not fully comprehend that it is not “surprising” that an inverse method that follows the generative assumptions of the forward problem works well, as no previous method has been available at all. Moreover, we argue that it is precisely this conceptual clarity that makes our method attractive and widely applicable. The better estimation mentioned in the review would not be possible with previous methods, as we show in our experiments.
>
> The disambiguation between uncertainty and costs does not follow trivially but stems from using an inference method that accurately incorporates the sensing and acting uncertainties of the agent being modeled. If there are previously published methods that have all these properties, could you please point out IRL or IOC methods involving partial observability of the state, that can distinguish between epistemic and pragmatic actions, and allow for unobserved action signals?

---

### Author Rebuttal · Authors · 2023-08-09

We would like to thank all reviewers for their generally positive reviews. In light of some of the questions and comments, we would like to clarify the current state of inverse reinforcement learning (IRL), inverse optimal control (IOC) when applied to human or animal behavior. Looking at the publications at Neurips from recent years, IRL methods have been published that are targeted at inferring animal behavior, which, however completely exclude perception (partial observability due to sensory uncertainty from the perspective of the agent), assume the actions to be fully observed (the action observed by the researcher may not be identical to the planned and intended movement and the control signal, e.g. acceleration, is not directly measure, e.g. position instead). Thus, this IRL work omits explicitly modeling perception, i.e. sensory uncertainty. Accordingly, this does not allow to estimate an observation function or perceptual noise and there is accordingly not even a notion of epistemic versus pragmatic actions.

In recent IOC work, methods typically work in deterministic, fully-observable settings and assume the agent’s action signals to be observed (i.e. most problems implemented in OpenAI gym and comparable frameworks). There is IOC work that explicitly assumes partial observability from the perspective of the agent, i.e. involving sensory noise. However, to the best of our knowledge, there is no current method that can accommodate the setting considered here, i.e. problems that are stochastic, partially observable (both the partial observability introduced by perception from the perspective of the agent as well as the partial observability of the true intended actions by the agent from the perspective of the researcher), non-Gaussian noise, which is modeled by passing Gaussian noise through the nonlinear functions describing dynamics and observations.

We are motivated by problems relevant to neuroscience, motor control, cognitive science, and psychology, where biological systems are often modeled as having noisy sensory and motor systems, resulting in stochastic, partially-observable problems with unobserved action signals. For more naturalistic, sequential tasks, that are the frontier in current research, intrinsic beliefs and subjective costs are unknown. Moreover, when considering limb kinematics or muscle activations, dynamics are nonlinear, but usually well characterized, e.g. derived from first principles such as kinematics or measured empirically in separate experiments. Sensory uncertainty also involves nonlinearities, e.g. visual angles. Additionally, sensory noise and action noise are overwhelmingly non-Gaussian but well characterized, e.g. signal-dependent noise.

This complicates the inverse optimal control problem tremendously. Here, we present a method that is applicable in this setting. To our knowledge, it is the first such method applicable to this setting and able to distinguish between epistemic and pragmatic actions. Therefore, we see broad applicability and utility in the present work from neuroscience to motor control and beyond.

---

### Decision · Program_Chairs · 2023-09-21

**Decision:**

Accept (poster)

**Comment:**

This paper presents a new formalization of inverse optimal control in POMDPs. Using local linearization, a closed form approximation of the likelihood function for state trajectories is constructed to subsequently yield maximum likelihood estimates. This approach extends iterative linear quadratic Gaussian (iLQG) and Maximum causal entropy (MCE) reinforcement learning, introducing local mineralization to achieve tractable likelihood. The method is evaluated through simulations. The paper received overall mixed reviews. The reviewers appreciated the clarity of the paper, the novelty of the method, and the design of the experiments. One reviewer however think that the evaluation lacks sufficient quantitative and qualitative analysis. This opinion does not seem to be shared by other reviewers.